# Transcription factor network analysis identifies REST/NRSF as an intrinsic regulator of CNS regeneration in mice

Yuyan Cheng[1,12], Yuqin Yin[2,3,4,12], Alice Zhang[1], Alexander M. Bernstein[5], Riki Kawaguchi[1,6], Kun Gao[1], Kyra Potter[1], Hui-Ya Gilbert[2], Yan Ao[5], Jing Ou[1], Catherine J. Fricano-Kugler[1], Jeffrey L. Goldberg[7], Zhigang He [3,8], Clifford J. Woolf [3,8], Michael V. Sofroniew [5,13], Larry I. Benowitz [2,3,4,9,13] & Daniel H. Geschwind [1,6,10,11,13]

The inability of neurons to regenerate long axons within the CNS is a major impediment to improving outcome after spinal cord injury, stroke, and other CNS insults. Recent advances have uncovered an intrinsic program that involves coordinate regulation by multiple transcription factors that can be manipulated to enhance growth in the peripheral nervous system. Here, we use a systems genomics approach to characterize regulatory relationships of regeneration-associated transcription factors, identifying RE1-Silencing Transcription Factor (REST; Neuron-Restrictive Silencer Factor, NRSF) as a predicted upstream suppressor of a pro-regenerative gene program associated with axon regeneration in the CNS. We validate our predictions using multiple paradigms, showing that mature mice bearing cell type-specific deletions of REST or expressing dominant-negative mutant REST show improved regeneration of the corticospinal tract and optic nerve after spinal cord injury and optic nerve crush, which is accompanied by upregulation of regeneration-associated genes in cortical motor neurons and retinal ganglion cells, respectively. These analyses identify a role for REST as an upstream suppressor of the intrinsic regenerative program in the CNS and demonstrate the utility of a systems biology approach involving integrative genomics and bioinformatics to prioritize hypotheses relevant to CNS repair.

[1]Program in Neurogenetics, Department of Neurology, David Geffen School of Medicine, University of California, Los Angeles, Los Angeles, CA 90095, USA. [2]Department of Neurosurgery, Boston Children's Hospital, Boston, MA 02115, USA. [3]F.M. Kirby Neurobiology Center, Boston Children's Hospital, Boston, MA 02115, USA. [4]Department of Neurosurgery, Harvard Medical School, Boston, MA 02115, USA. [5]Department of Neurobiology, David Geffen School of Medicine, University of California, Los Angeles, Los Angeles, CA 90095, USA. [6]Department of Psychiatry, Semel Institute, University of California, Los Angeles, Los Angeles, CA 90095, USA. [7]Byers Eye Institute and Wu Tsai Neuroscience Institute, Stanford University, Palo Alto, CA 94305, USA. [8]Department of Neurology, Harvard Medical School, Boston, MA 02115, USA. [9]Department of Ophthalmology, Harvard Medical School, Boston, MA 02115, USA. [10]Department of Human Genetics, David Geffen School of Medicine, University of California, Los Angeles, Los Angeles, CA 90095, USA. [11]Semel Institute for Neuroscience and Human Behavior, David Geffen School of Medicine, University of California, Los Angeles, Los Angeles, CA 90095, USA. [12]These authors contributed equally: Yuyan Cheng, Yuqin Yin. [13]These authors jointly supervised this work: Michael V. Sofroniew, Larry I. Benowitz, Daniel H. Geschwind. ✉e-mail: larry.benowitz@childrens.harvard.edu; dhg@mednet.ucla.edu

Injured axons in the adult mammalian central nervous system (CNS) generally cannot regenerate over long distances, limiting functional recovery from CNS injury[1]. Potential mechanisms underlying regenerative failure in the mature CNS include a lack of an intrinsic ability to activate genes and pathways required for axon regrowth after injury; the presence of extrinsic growth-repulsive factors associated with certain extracellular matrix molecules, myelin debris, or fibrotic tissue; and limited availability of appropriate growth factors. Strategies to neutralize or attenuate key cell-extrinsic inhibitors of axon growth have limited effects on regeneration[2], though their impact is strongly enhanced by co-activating neurons' intrinsic growth state[3]. Deleting PTEN, a cell-intrinsic suppressor of axon growth, induces appreciable axon regeneration, and when combined with either CNTF plus SOCS3 deletion, or with inflammation-associated factors plus cAMP, enables a percentage of retinal ganglion cells to regrow axons the full length of the optic nerve[4–6]. Nonetheless, more work is needed to identify key regulators of axon regeneration in the CNS, including transcription factors that act as master switches of the regenerative program.

Unlike their CNS counterparts, peripheral sensory and motor neurons spontaneously display potent growth in response to peripheral axonal injury, which is accompanied by activation of key regeneration-associated genes (RAGs)[7] that we found to act as a coordinated network to promote growth[8]. Expression of this RAG network is predicted to be regulated by a core group of TFs during peripheral nerve regeneration[8]. This hypothesis is supported by the findings that manipulating individual TFs at the core of this network, such as STAT3[9] and Sox11[10] result in varying amounts of CNS axon growth. The effects of TFs on their target pathways are dynamic, combinatorial, and form tiered regulatory networks, requiring tight control in timing, dosage, and the context of each TF involved[11–13]. The complexity of recapitulating coordinated TF regulatory events may limit the effectiveness of single gain- or loss-of-function experiments to determine contributions of individual TFs within a complex network[14]. Alternatively, illuminating the hierarchical transcriptional network architecture from gene expression datasets provides an efficient means to identify key upstream regulators of various biological processes[15], for example, pluripotency[16]. One model of TF networks originally used by the ENCODE consortium relies on a 3-level pyramid-like structure, with a small number of TFs at the top-level that function as 'master' regulators, driving expression of most of the other mid- and bottom level TFs that directly or indirectly regulate the expression of their target genes[13,17].

Here, we integrated multiple existing and newly generated datasets to characterize hierarchical TF interactions, so as to identify potential upstream regulators associated with the intrinsic axon regeneration state (Fig. 1a). By comparing gene expression in non-permissive states, such as the injured CNS, to the permissive PNS or to the CNS that has been subjected to strong pro-regenerative treatments, we hypothesized that we could identify key upstream TFs driving intrinsic regeneration programs. We began with a mutual information-based network analysis approach to characterize the transcriptional regulatory network formed by regeneration-associated TFs[8] in multiple independent data sets. We identified a core subnetwork of five interconnected TFs, consisting of Jun, STAT3, Sox11, SMAD1, and ATF3, which is strikingly preserved across multiple PNS injury models and at different timescales[18–21]. Remarkably, we observe a similar multi-layer, highly inter-connected TF structure in CNS neurons following genetic and pharmacological treatments that enhance regeneration. In contrast, in the non-regenerating CNS at baseline[18,22,23], this regeneration-associated subnetwork and its hierarchical structure are dismantled, and candidate TFs adopt a less interconnected and less hierarchical structure.

Our analyses identified RE1-silencing transcription factor (REST;[24,25]), a widely studied regulator of neural development and neural-specific gene expression[24–26], as playing a potentially important role in suppressing CNS regeneration (Fig. 1a). Our findings suggested that REST acts as a potential upstream transcriptional repressor, limiting the interactions of the core regenerative TFs to drive the expression of RAGs and the intrinsic growth capacity of CNS neurons (Fig. 1b). This hypothesis was supported by transcriptomic analysis of REST-depleted, CNS-injured neurons, which displayed enhanced expression of a regeneration-associated gene network, driven by several core TFs known to promote regeneration. To further validate our bio-informatic predictions, we investigated the effects of counteracting REST on regeneration in two different models of CNS injury in vivo—optic nerve crush and complete spinal cord injury (SCI)—via conditional depletion or functional inactivation of REST in retinal ganglion cells (RGCs) and corticospinal tract (CST) projection neurons (Fig. 1c), respectively. In both cases, counteracting REST resulted in increased regeneration. These findings illustrate how a multi-step systems- biological analysis coupled with substantial in vitro and in vivo experimental validation facilitates for discovery of drivers of CNS repair, and implicate REST as an important regulator of CNS axon regeneration.

## Results

### Bio-informatic analysis identifies REST as a potential upstream repressor of a regeneration-associated network

To determine which of the previously identified pro-regenerative TFs[8] are essential drivers of neurons' intrinsic growth program, we characterized the regulatory network among these TFs to define their directional and hierarchical relationships using the step-wise approach summarized in Fig. 2a. To infer directionality of each pair of TFs, we applied the Algorithm for Reconstruction of Accurate Cellular Networks (ARACNe), a mutual-information (MI) based algorithm for reverse-engineering a transcriptional regulatory network from gene expression datasets[27]. ARACNe connects two genes only if there is an irreducible statistical dependency in their expression. These connections likely represent direct regulatory interactions mediated by a TF binding to its target genes, and thus can be used to predict the TF network and the transcriptional targets[27] (Fig. 2a; Methods). These predictions have been extensively validated by experimental analysis, such as chromatin immunoprecipitation sequencing (ChIP-Seq), a method to identify physical TF-target binding, or by examining expression changes of target genes led by gain- or loss- of function of the regulatory TFs[28–32].

In an ARACNe-constructed transcriptional regulatory network, a TF is either predicted to have a positive edge with its target genes (i.e., activator of expression; MI (+)) when their expression patterns are positively correlated, or negative edge (i.e., repressor of expression, MI (-)) if the TF displays opposite transcriptional changes from its targets (Fig. 2a, step 1). We subsequently validated the initial bio-informatic predictions of edge directionality by compiling direct biochemical evidence of physical TF-target binding observed by multiple ChIP-Seq or ChIP-ChIP databases[33,34], leading to a high-confidence, directed TF regulatory network supported by experimental evidence (Fig. 2a, step 2). Lastly, the hierarchical structure of the directed TF network was defined by *vertex-sort*, a network decomposition algorithm[17], which elucidates the topological ordering of members by their connectivity in a directed network (Fig. 2a, step 3).

Because TF binding is a dynamic process that may change over time, we analyzed 9 high-density time-series gene expression profiles from injury models to build the networks, leveraging the chronological order of regulatory events. By applying our step-wise pipeline to 6 peripheral nerve and 3 spinal cord injury datasets (Fig. 2b; Methods), we sought to identify reproducible differences in transcriptional regulatory networks between regenerating PNS and non-regenerating CNS neurons following injury. We found that the candidate TFs regulate each other within complex, multi-layered networks, similar to TF network models defined by ENCODE (Fig. 2c;[13,35]). Across multiple

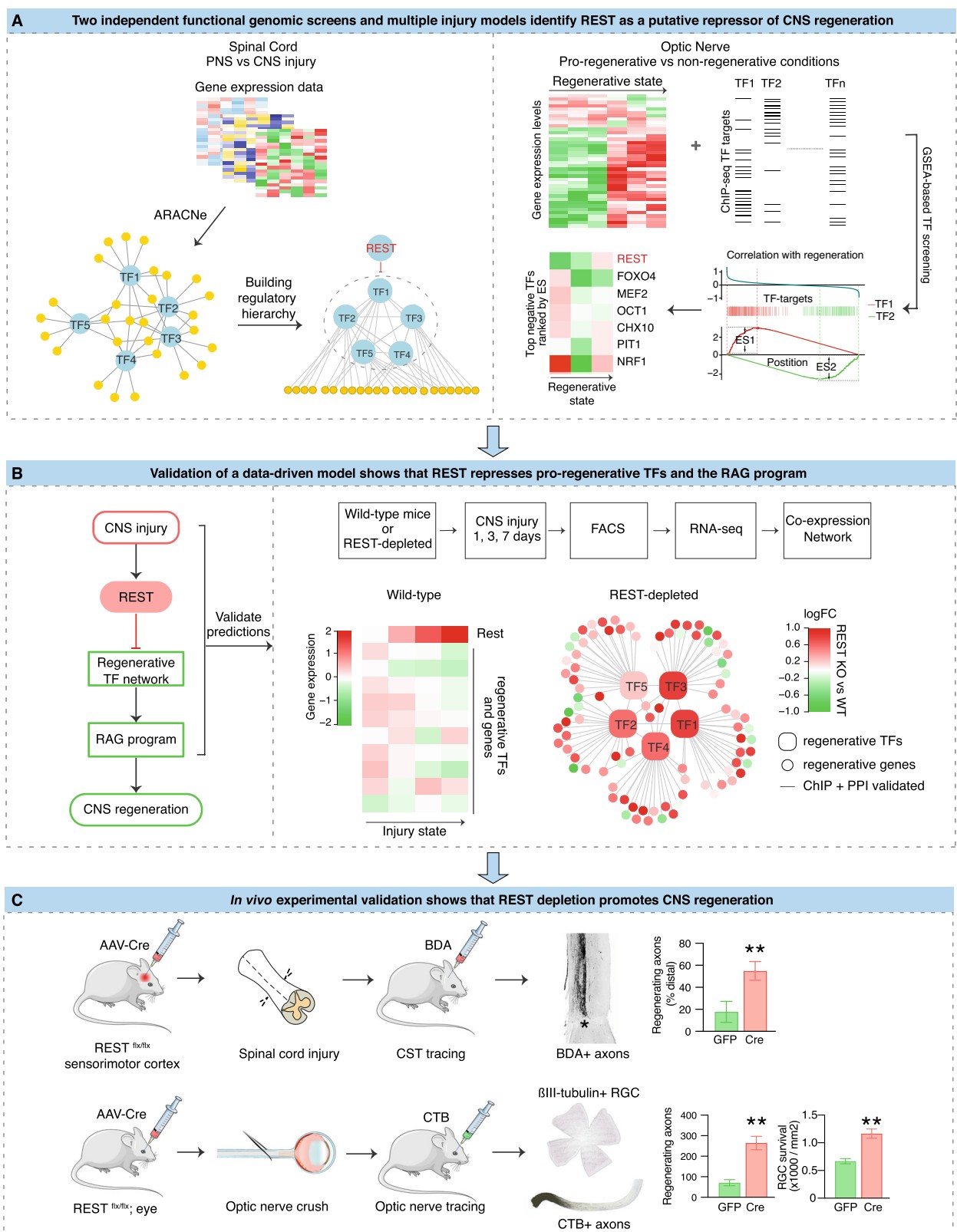

datasets in multiple PNS injury models collected in different laboratories at different timescales, we observed a remarkable preservation of a defined five TF subnetwork consisting of JUN, STAT3, SOX11, SMAD1, and ATF3 (Fig. 2c), all of which were consistently increased after PNS injury (Supplementary Fig. 1a, PNS1-3) and are required for peripheral nerve regeneration[8,36–40]. In contrast, in the CNS, this subnetwork appeared dismantled and adopted a simpler, bi-layered, and less inter-connected network in the case of CNS injury (Fig. 2d). The key TFs within this subnetwork are either not induced by injury (Supplementary Fig. 1a, CNS1 and 2), or are transiently up-regulated, but quickly down-regulated (Supplementary Fig. 1a, CNS3).

To quantitate the network differences observed in the PNS and CNS injury models, we first calculated the global and local clustering coefficient, the former indicating the global network connectivity, and

**Fig. 1 | Schematic diagram summarizing the overall experimental flow integrating iterative bio-informatics and experimental validation.** Multiple independent functional genomics analyses of distinct injury models were analyzed to computationally identify upstream TFs associated with CNS regeneration. In the first set of analysis (**a**, left), we performed a mutual information-based network analysis using ARACNe to characterize the transcriptional regulatory network formed by regeneration-associated TFs in multiple independent data sets from spinal cord and peripheral nerve injury. The hierarchical structure of the TF regulatory network was further characterized so as to identify potential upstream regulators. This step-wise analysis predicted REST, a transcriptional repressor, as an upstream negative regulator inhibiting the core pro-regenerative TFs to drive the expression of regeneration-associated genes (RAGs). In parallel (**a**, right), we performed an additional TF screen in another CNS tissue, optic nerve, under pro-growth and native conditions to identify TF regulators of regeneration. Among the -1000 TF-target gene sets tested via Gene Set Enrichment Analysis, REST was ranked as the top negative regulator of the RGC regeneration state-associated gene set. Multiple independent bio-informatic analyses of external data sets confirmed and converged on our model **b**, by which REST is activated by CNS injury and acts as a potential upstream negative regulator of the core regenerative TFs. To test this, we performed gene expression analysis in the injured CNS with REST and after REST depletion, showing REST increases following CNS injury, while the core pro-regenerative TFs and genes remain suppressed. Depleting REST activates a core molecular program driven by a tightly controlled TF network similar to the one activated during regeneration. These results predicted that REST depletion would improve regeneration, which we directly tested in two well-established models of regeneration in vivo **c**, confirming REST's functional effect as a suppressor of regeneration. In the case of optic nerve injury, REST depletion or inhibition enhanced both RGC regeneration and survival. These analyses identify a role for REST as an upstream suppressor of the intrinsic regenerative program in the CNS and demonstrate the power of a systems biology approach involving integrative genomics analysis to predict key regulators of CNS repair.

the latter summarizing the local connectivity of each TF node. We observed a higher global clustering coefficient across all PNS networks compared to the CNS ones (Supplementary Fig. 2a, global CC). Likewise, the local connectivity of the five regenerative TFs is consistently higher in the PNS than in the CNS (Supplementary Fig. 2a, local CC). Next, we investigated the occurrence of network motifs, the fundamental building blocks of diverse classes of network architectures[15], within each PNS or CNS TF network. We observed the most enriched motif structure within the PNS networks was the well-studied feed-forward loop (FFL), whereas the CNS networks lack enrichment or depletion in most of the motif structures containing three nodes (Supplementary Fig. 2b), likely due to their simpler, bi-layered network structure (Fig. 2d). Lastly, we calculated the similarity of each key TF's regulon across PNS and CNS datasets. The five regenerative TFs, ATF3, JUN, SOX11, and SMAD1 bear the most correlated regulatory relationships with each other across multiple PNS injury datasets (Supplementary Fig. 2c PNS vs PNS), further supporting their preserved network structure in the PNS (Fig. 2c). By contrast, there is little correlation in the regulatory interactions of the core TFs between PNS and CNS injury datasets (Supplementary Fig. 2c, CNS vs PNS). Altogether, these findings support a more inter-connected, reproducible network structure of regeneration-associated TFs in the PNS than in the CNS counterparts.

Remarkably, we observed that two TFs, REST and CTCF, appear to interact with top-tier TFs in the CNS, but not in the PNS, network, and are predicted to inhibit other top-tier TFs. *Rest* mRNA levels did not change after PNS injury (Supplementary Fig. 1a, PNS1-3), but were increased by CNS injury when other key regenerative TFs begin to be repressed (Supplementary Fig. 1a, CNS1-3). We did not observe changes of *Ctcf* expression levels following PNS or CNS injury. We therefore hypothesized that REST, which appears at the apex of a less inter-connected TF network, is a potential upstream transcriptional repressor of the core TF network in the non-regenerating CNS, thus limiting interactions among the core TFs to drive the expression of regeneration-associated genes and to activate the intrinsic growth program of CNS neurons.

## REST deletion in CNS-injured neurons increases expression of growth-related genes and pathways

If REST were indeed an upstream repressor as predicted, its depletion in CNS neurons would be expected to release the transcriptional brake of pro-regenerative TFs and downstream genes, thereby increasing their expression. To test this hypothesis, we performed RNA-seq on REST-depleted sensorimotor cortical neurons that give rise to the corticospinal tract (CST) axons that course through the spinal cord. The CST is essential for controlling voluntary motor movements, and the failure of CST axons to regenerate is a major impediment to improving outcome after spinal cord injuries. To induce neuron-

specific REST depletion, we injected adeno-associated virus expressing Cre recombinase, or GFP as a control, under a synapsin promoter (AAV-Syn-Cre and AAV-Syn-GFP) in the sensorimotor cortex of mice with homozygous conditional REST alleles and a TdTomato reporter (REST^flx/flx; STOP^flx/flx TdTomato mice; Methods). REST knock-out (cKO) was confirmed by tdTomato expression in the cortical area of REST^flx/flx; STOP^flx/flx TdTomato mice injected with AAV-Syn-Cre. No TdTomato was observed in control mice receiving AAV-Syn-GFP. We then performed an anatomically complete spinal cord crush injury (SCI) at thoracic level 10 (T10) to avoid spontaneous axon regeneration (i.e., circuit reorganization) that can occur after incomplete injury[41]. Following sham or T10 SCI, neurons expressing GFP (wild-type) or tdTomato (REST cKO) were FACS-sorted at multiple time points post injury for RNA sequencing (Fig. 3a; Methods). We then analyzed transcriptional differences in response to SCI and REST depletion at both the individual gene expression level and co-expression network level. Integrating network-level analysis complements analysis of differential expression by reducing the dimensionality of a large transcriptomic dataset and helps to identify clusters of genes sharing expression patterns and biological functions[42].

We first examined differentially expressed genes in response to injury alone (Supplementary Fig. 3a, b). In wild-type neurons expressing AAV-GFP, SCI resulted in the up-regulation of genes involved in both injury- and regeneration-associated processes at day 1, including apoptosis, neuron projection, cell adhesion, and axon extension (Supplementary Fig. 3c)[22,23]. At days 3 and 7 post-injury, however, the up-regulated genes were predominantly associated with injury-relevant pathways involved in oxidative stress, and receptors or channels that increase neural excitability (Supplementary Fig. 3c)[22,23]. REST expression levels increased in sensorimotor cortex neurons at 3 and 7 days post-injury (Fig. 3b, AAV-Syn-GFP) in parallel with the expression of injury-relevant gene expression patterns.

The timing of REST expression subsequent to the early (but aborted) regeneration pathways, and prior to more subacute injury-related pathways, is consistent with REST potentially repressing regeneration-associated genes. To test this hypothesis, we compared gene expression responses in sorted, purified, sensorimotor cortex neurons with or without REST deletion at multiple time points post SCI. At early time points, only a few genes were responsive to REST deletion, whereas far more DEGs were identified at 7 days following injury (day 0 or sham condition [up-regulated: 40; down-regulated: 65]; day 3 [up-regulated: 39; down-regulated: 106]; day 7 [up-regulated: 351; down-regulated: 517]), consistent with the observed time-dependent increase of REST following SCI. A gene ontology analysis showed that up-regulated genes at 7 days following injury with REST deletion are involved in regulation of neural transmission, neuron projection, and neurite growth or patterning, whereas the down-regulated genes are associated with protein translation, mRNA

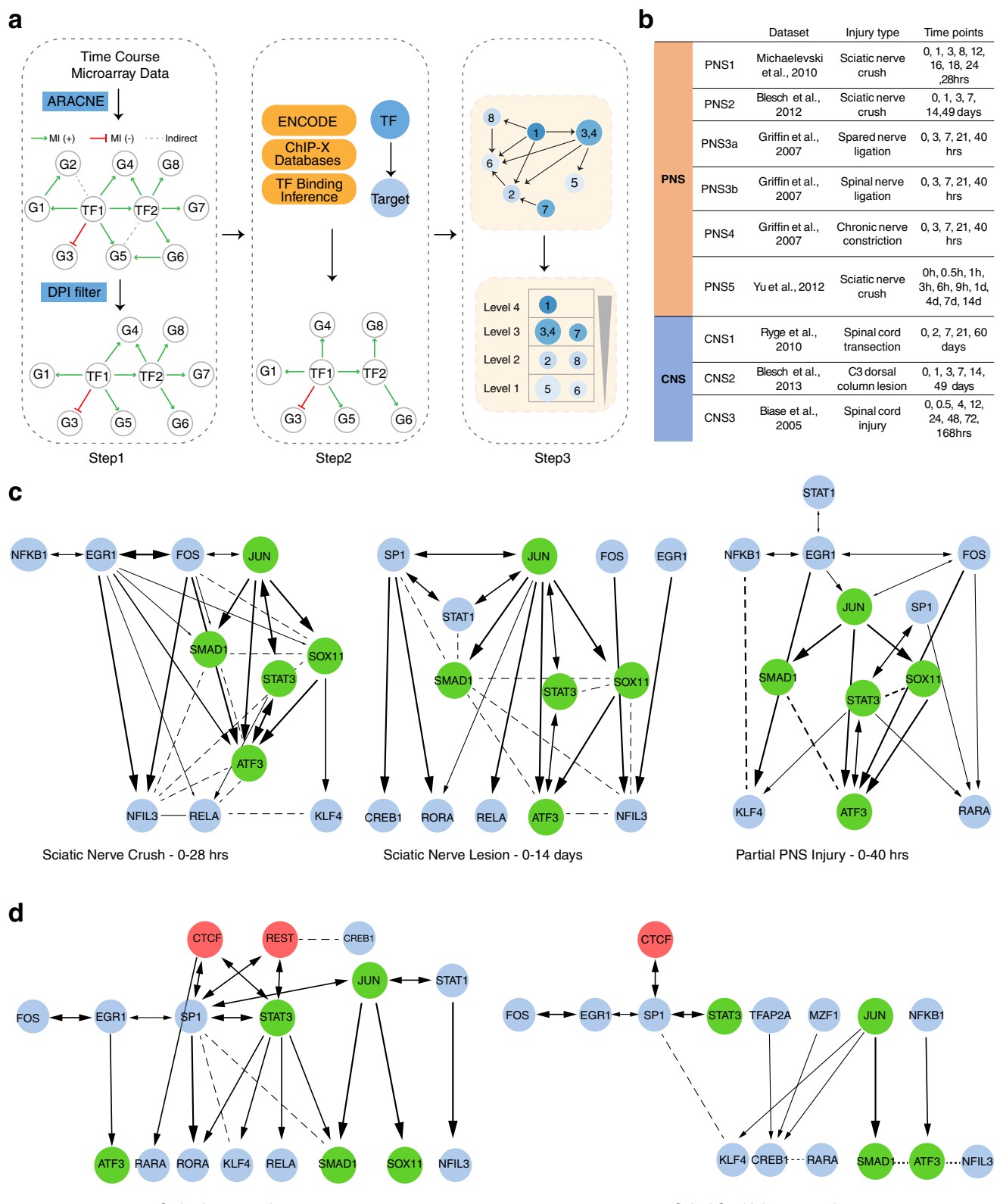

processing, and cell cycle (Supplementary Fig. 4a). Remarkably, expression levels of the core five peripheral axon regeneration-associated TF network genes (*Jun, Smad1, Sox11, Stat3, and Atf3*) (Fig. 2) were all up-regulated in REST-depleted neurons (Fig. 3b), with *Jun* and *Atf3* significantly increased at day 3 post SCI, and *Smad1, Sox11, Stat3* significantly increased by day 7. Notably, other TFs or known genes in the RAG program that we previously characterized in the PNS[8]

were also increased by REST depletion (Supplementary Fig. 4b), including immediate early genes induced by peripheral injury (*Egr1*), growth-associated proteins (*Gap43, Cap23*), molecules involved in vesicle and cytoskeletal transport (*Vav2, Syt4*), cell proliferation (*Pcna*), cAMP signaling (*Rapgef4/Epac2*) and p38 MAPK signaling (*Atf2, MApkapk2*). It is also possible that REST inhibition improves regeneration by abrogating the transcription of other TFs that are known to limit

**Fig. 2 | Characterizing regeneration-associated transcriptional regulatory network. a** Schematic diagram illustrating step-wise approaches employed to infer hierarchical TF regulatory networks from **b** time-course microarray datasets. Step 1: First, ARACNe was applied to each dataset to find TF-target pairs that display correlated transcriptional responses by measuring mutual information (MI) of their mRNA expression profiles (Methods). The sign (±) of MI scores indicates the predicted mode of action based on the Pearson's correlation between the TF and its targets. A positive MI suggests activation of this TF on its targets, while a negative MI score suggests repression. All non-significant associations were removed by permutation analysis. Second, ARACNe eliminates indirect interactions, such as two genes connected by intermediate steps, through applying a well-known property of MI called data-processing inequality (DPI). Step 2: To determine the

direction of regulation between each TF interactions, ChIP-datasets from ENCODE and previously published ChIP-ChIP and ChIP-seq datasets were integrated to compile a list of all observed physical TF-target binding interactions. Step 3: To identify the hierarchical structure within directed TF networks, we used graph-theoretical algorithms to determine precise topological ordering of directed networks based on the number of connections that start from or end at each TF, indicating whether a TF is more regulating or more regulated. **c, d** Representative regulatory networks inferred from microarrays following peripheral nerve injury **c** and CNS injury **d**. Each node represents one of the 21 regeneration-promoting TFs if a connection exists. The thickness of each line indicates the MI between the TFs it connects. A directional arrow is drawn if there is direct physical evidence of the TF binding its target TF's promoter.

---

CNS regeneration, such as *Pten, Socs3, and Klf4*. We did not observe a significant change in expression of these well-defined repressors of regeneration, however (Supplementary Fig. 4c), suggesting that REST is likely acting independently of these known repressive molecules. Overall, these findings indicate that REST is up-regulated by CNS injury (Fig. 3b) and that it transcriptionally represses its canonical neuronal target genes as well as the regeneration-associated TFs, as predicted by our bio-informatic analysis (Fig. 2 and Supplementary Fig. 1).

## REST deletion enhances a co-expression network associated with regeneration

Next, we used Weighted Gene Co-expression Network Analysis (WGCNA)[42,43] to identify network-level changes regulated by REST. WGCNA is less computationally intensive than ARACNe and identifies larger modules of highly co-expressed genes based on correlation[42,43], rather than mutual information[27]. In addition, we previously showed that WGCNA modules could be further integrated with experimentally validated protein-protein interactions (PPI) to identify protein-level signaling pathways represented by gene networks[8]. This would not only provide independent support of the relationship inferred by RNA co-expression, but also point to important PPI pathways for potential therapeutic intervention.

We performed WGCNA on our cortical neuron RNA-seq data, comparing AAV-Syn-GFP (wild-type) *vs.* AAV-Syn-Cre (REST-depleted) at 1, 3, and 7 days following SCI (Methods; Supplementary Fig. 5a–c). Based on the correlation of the first principal component of a module, called the module eigengene, with time-dependent changes after injury, we found five modules significantly altered by REST deletion: RESTUP1, RESTUP2, and RESTUP3, which were all up-regulated by REST deletion, and RESTDOWN1 and RESTDOWN2, which were down-regulated (Fig. 3c lower panel, Fig. 3d). To determine which of these gene modules altered by REST deletion are associated with regeneration, we performed an enrichment analysis between each module and the core RAG co-expression module, which we previously identified to be activated during peripheral nerve regeneration and enriched for regeneration-associated pathways in multiple independent data sets[8]. This analysis found that the up-regulated modules RESTUP1 and RESTUP3 significantly overlap with a core RAG co-expression module that occurs in the PNS, but not in the CNS under baseline conditions (Fig. 3c, upper panel).

Among the pathways associated with this core RAG module, the RESTUP3 module is enriched with cAMP-mediated, Ephrin-, PKA-, TGFβ-, GPCR- and MAPK signaling, while the RESTUP1 module is modestly enriched with integrin-, chemokine-, and HMGB1 signaling pathways (Fig. 3e). To extend this analysis to the protein level, we evaluated the overlap between PPIs from co-expressed genes in RESTUP1 or RESTUP3 and the regeneration-associated PPIs from the RAG module. We found that PPIs from RESTUP3 and RESTUP1 are enriched for very similar regeneration-associated pathways shown by gene-level overlap analysis, which are linked by members of the core TF regulatory network activated in the regenerating PNS (Fig. 3f,

Supplementary Data 2), including Jun, SMAD1, STAT3 and ATF3 (Fig. 2b, Supplementary Fig. 1a). These core regenerative TFs also appear as module hubs in the PPI network of the RESTUP3 module (Supplementary Fig. 5d). Further GO analysis of general biological pathways represented by these modules show that the RESTUP3 module is associated with neuronal projection, metabolism, or synaptic transmission (Supplementary Fig. 5e). These analyses support a model whereby inhibition of REST activates a core molecular program driven by a tightly controlled TF network similar to the one activated during peripheral nerve regeneration, along with other complementary pathways, to enable the subsequent regenerative process (Supplementary Fig. 5f).

## REST is a transcriptional repressor negatively correlated with the regenerative state of retinal ganglion cells

To assess the potential generalizability of the bio-informatic predictions derived from spinal cord and peripheral nerve injury models, we extended the same TF regulatory network analysis to another CNS neuronal population, injured retinal ganglion cells (RGCs). RGCs extend axons through the optic nerve, conveying diverse visual features to the lateral geniculate nucleus, superior colliculus, and other relay centers in the di- and mesencephalon, and are a widely studied example of CNS neurons that normally exhibit little or no regenerative potential[1]. However, although mature RGCs fail to regenerate axons beyond the site of optic nerve injury and soon begin to die, varying degrees of regeneration can be induced by treatments that include growth factors associated with intraocular inflammation, CNTF gene therapy, deletion of cell-intrinsic suppressors of axon growth (of which PTEN deletion is the single most effective), zinc chelation, physiological activity, counteracting cell-extrinsic suppressors of axon growth, chemical activation of the regenerative gene program and, most effectively, by combining two or more of these treatments[3,4,6,8].

From our initial bio-informatic predictions comparing PNS and CNS injured tissues, we hypothesized that the disrupted TF network in injured, non-growing RGCs, like the CNS-injured spinal cord tissues (Fig. 2d), would regain substantial connectivity in RGCs treated to attain a regenerative state. Using mice that express cyan-fluorescent protein (CFP) in RGCs[44], we induced robust axon regeneration by combining a strong genetic pro-regenerative manipulation, RGC-selective PTEN knock-down (AAV2-shPten.mCherry; Methods;[45]), with intraocular injection of the neutrophil-derived growth factor onco-modulin (Ocm;[46,47]) and the non-hydrolyzable, membrane-permeable cAMP analog CPT-cAMP (a co-factor of Ocm) immediately after nerve injury. This combination provides one of the strongest regenerative responses described to date (Fig. 4a), while avoiding complications that might be introduced by intraocular inflammation[5]. Controls received an intraocular injection of AAV2 expressing shLuciferase.mCherry 2 weeks before surgery and saline immediately afterwards and did not exhibit axon regeneration (Fig. 4a, b; see Methods). We dissected retinas and FACS-sorted from non-regenerating, control RGCs or from RGCs exposed to the pro-regenerative combinatorial

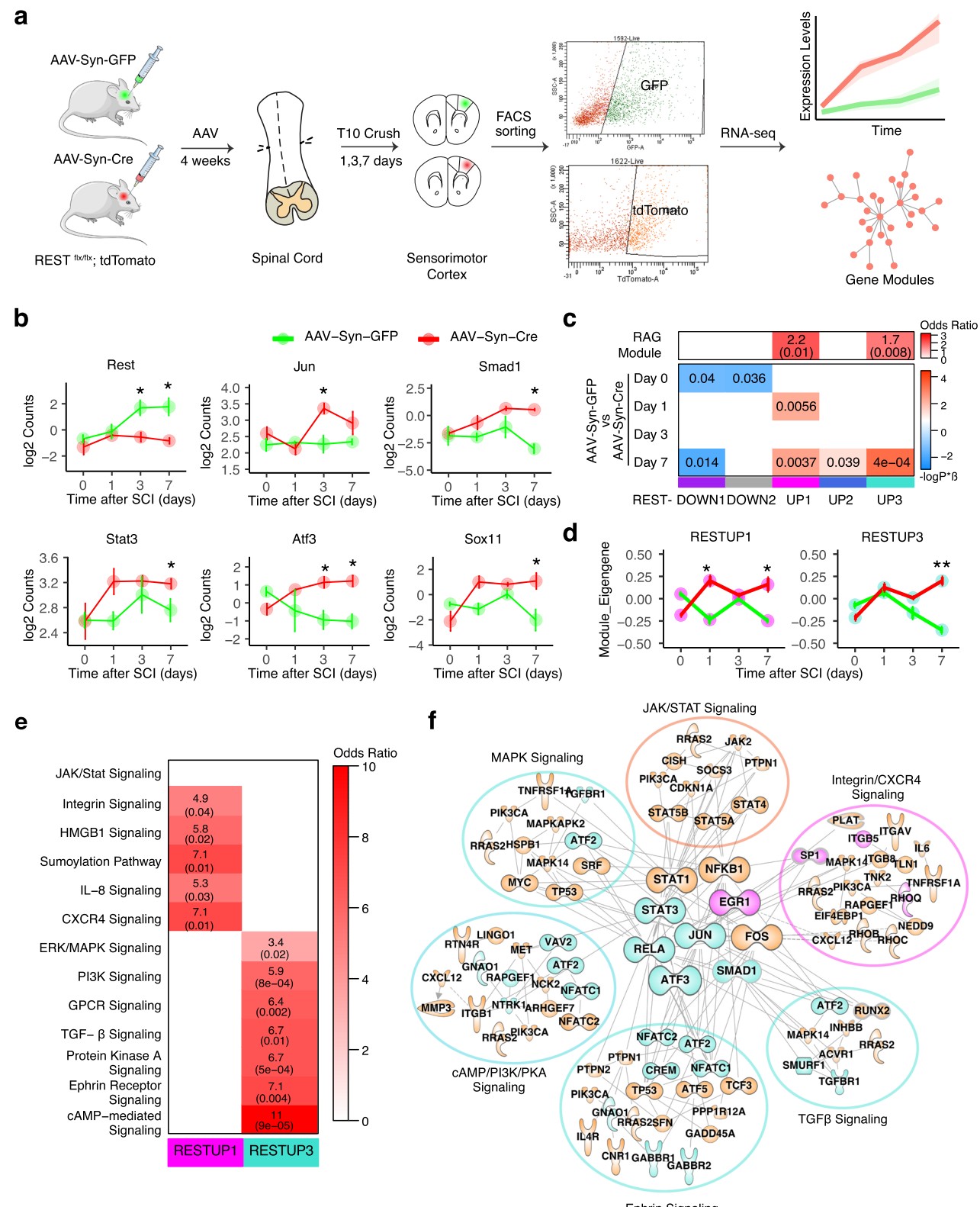

treatment 1, 3, or 5 days after optic nerve crush injury, followed by transcriptomic analysis via RNA-seq (Fig. 4c; Methods).

To quantitatively determine a TFs' association with RGCs' regeneration state, we first performed gene set enrichment analysis (GSEA) to compare a gene expression signature correlated with the RGC axon regenerative state against 'tag gene sets' with known binding sites for TFs[48]. GSEA returns an enrichment score (ES) of this comparison to determine whether the gene set represented by

regeneration-associated genes is enriched in targets of any TFs and if it is a positive or negative regulator of the genes associated with regeneration phenotype (Fig. 4d;[49]). Among the ~1000 TF-target gene sets tested, REST is ranked as the top negative regulator of the RGC regeneration state-associated gene set at day 1 following injury, which is attenuated on days 3 and 5 after injury (Fig. 4d). These results are consistent with REST being an early, upstream element in the regulatory cascade.

**Fig. 3 | REST deletion in injured cortical neurons enhances expression of regeneration-associated genes and pathways. a** Overview of RNA-seq of FACS-sorted cortical neurons expressing AAV-Syn-GFP (wild-type) or AAV-Syn-Cre (REST cKO) after a complete crush injury at thoracic spinal cord level 10 (T10). Transcriptional differences in response to SCI and REST depletion were analyzed for differential expression and changes in co-expression newtworsk. **b** Expression levels of *Jun, Smad1, Sox11, Stat3, Atf3, and Rest*. Values are mean log2 Counts ± SEM from RNA-seq data. *n* = 3 mice in each condition. Asterisks denotes FDR-corrected, two-tail *P* < 0.1 compared to AAV-Syn-GFP at each time point. Exact P values and "n" used in each condition are included in Supplementary Data 1. **c** WGCNA modules with significant correlations to treatments (bottom panel) and over-representation of regeneration-associated genes (RAGs)[20] within each module (upper panel). In the correlation heatmap, colors indicate −sign(correlation coefficient)*(log10 p-value). In the enrichment heatmap, numbers shown are odds ratio indicating the possibility of enrichment, with the hypergeometric p-value in parenthesis. **d** Trajectory of the RESTUP1 and RESTUP3 module eigengenes (MEs) across different time points after SCI in AAV-Syn-GFP (green) and AAV-Syn-CRE expressing (red) neurons. Values are mean MEs ± SEM; Asterisks denote statistical significance assessed by ANOVA model with Tukey's post-hoc test: *$p < 0.05$, **$p < 0.01$ compared to AAV-Syn-GFP. Exact P values are shown in c. **e** Over-representation (hypergeometric test) of subsets of RAGs in RESTUP1 and RESTUP3. These subsets of RAGs were derived from GO analysis. **f** Protein–protein interaction (PPI) network represented by genes in the RESTUP1, RESTUP3 and RAG modules. Each node represents a molecule from the RAG module, colored by orange, while edge represents an experiment-supported PPI between two nodes. Molecules that also appear in RESTUP1 are colored in magenta, while molecules appearing in RESTUP3 are colored in turquoise. The core transcription factors are placed in the center.

We next performed a complementary analysis using the same ARACNe-based pipeline as used in our initial analysis of published PNS and CNS microarray datasets to construct a data-driven hierarchical network of the regenerative TFs within this new RNA-seq dataset. Similar to CNS injured tissues in the first analysis (Fig. 2d), non-regenerative RGCs with control treatment adopt a simpler, less inter-connected, and less structured TF network. This unsupervised analysis again predicted that REST appears at the top of the non-regenerating network (Fig. 4e, Control), and is negatively correlated with other lower-layer TFs (Fig. 4f, Control). By contrast, the strong pro-regenerative treatment re-established a more complex, multi-layered network with higher connectivity (Fig. 4e, global clustering coefficient in Control = 0.25, versus the pro-regenerative treatment = 0.54), in which REST is dissociated and the key regenerative TFs (*Atf3, Jun, Sox11, Stat3*) are more connected (Fig. 4f), as in the microarray data from PNS (Fig. 2). Other commonly used statistics for network connectivity such as local clustering coefficient, between-ness centrality, and in- and out-degree (Methods), further revealed substantially higher connectivity for the RAG TFs in the regenerating versus non-regenerating group (Supplementary Fig. 6a). These results from independent datasets and different neural systems further support the original bio-informatic predictions that neurons displaying regenerative potential are associated with a highly inter-connected, structured TF-regulatory network. Further, these analyses (e.g., Figs. 2 and 4) show that REST emerges as an inhibitory TF at the apex of a less-connected TF network in the non-regenerating CNS neurons, but is not associated with the highly interacting TF network seen when CNS neurons are in a regenerating state.

These multiple analyses of independent datasets suggested that REST is an upstream transcriptional repressor that potentially limits the interactions between lower-level TFs and the expression of regeneration-associated genes. One prediction of this model is that REST target genes should be enriched in RAGs and RAG-associated processes, parallel with the GSEA (Fig. 4d). We observed 630 transcriptional interactions with REST predicted by ARACNe, including 339 positively regulated (activated) genes and 321 negatively regulated (repressed) genes (Supplementary Fig. 6b, Supplementary Data 4; Methods). Enriched GO terms for genes predicted to be activated by REST include metabolic processes, response to endoplasmic reticulum (ER) stress, and RNA binding and transport (Supplementary Fig. 6c), whereas genes predicted to be repressed by REST are indeed implicated in processes or pathways associated with axon regeneration[50], including calcium ion transport, axon guidance, synaptogenesis, and CREB- and cAMP-mediated signaling (Supplementary Fig. 6c). The REST-repressed, regeneration-associated gene set was enriched with down-regulated genes at early stages (day 1), which were up-regulated in the later stages of regeneration (day 3 and 5) (Fig. 4g, GSEA; Supplementary Fig. 6d), suggesting a release of the transcriptional brake by REST on these genes.

## REST binding activities at regeneration-associated TFs and genes

We next characterized REST binding at the promoters of pro-regenerative TFs and genes via DNA footprinting analysis using the assay for transposase-accessible chromatin with sequencing (ATAC-Seq) data, which is generated in our other study (GSE184547[51]), from FACS−sorted RGCs at 1 and 3 days following optic nerve injury. This analysis is based on the knowledge that TF-bound DNA are protected from transposase cleavage during ATAC-seq (Fig. 5a). Matching protected DNA sequences, or DNA "footprints" to TF motif databases enables identification of direct TF binding sites, similar to, but with higher resolution than ChIP-seq[52,53]. We mapped DNA footprints across open chromatin regions[51] overlapping with REST's binding motifs. We identified a total of 9,214 DNA-footprints predicted to be bound by REST at either day 1 or day 3 following injury, but not in the uninjured condition (day 0; Methods; Supplementary Data 5). By focusing on the promoter footprints (±2 kb of a gene's transcription start site), we identified a total of 801 REST-targeted genes. Consistent with REST's activity as a transcriptional repressor, we observed decreases in both promoter accessibility and mRNA levels for most REST-targeted genes (Fig. 5b). These REST-target genes overlap with genes identified by a previous REST ChIP-seq in adult hippocampal neural stem cells (OR $_{(ChIP)}$ = 2.72, p $_{(ChIP)}$ = 1.2e-14, Fisher's exact test)[54]. REST-bound genes also overlap with the regeneration-associated gene sets predicted to be regulated by REST in the RNA-seq datasets generated from RGCs (Fig. 4g) and cortical neurons (Fig. 3, module RESTUP1 and 3) in different regenerating states (OR $_{(RGC)}$ = 5.75, p $_{(RGC)}$ = 2.9e-09; OR $_{(Cortical)}$ = 1.58, p $_{(Cortical)}$ = 1.2e-03). Among the genes directly bound by REST were the key regenerative TFs *Atf3, Stat3, Smad1*, and *Sox11*. Interestingly, REST binding activity around these genes, which is measured by scoring each footprint's signal and the surrounding chromatin accessibility, was increased at day 1 following CNS injury but decreased by day 3 (Fig. 5c). These results, together with the findings of increased expression of regenerative TFs upon REST inhibition (Fig. 3b; Fig. 8g–i), provide strong evidence that REST directly binds to and represses expression of TFs that would otherwise drive axon regeneration.

Altogether, multiple independent analyses of data from different sources that were focused on identifying key upstream TFs regulating CNS regeneration revealed REST to be a key transcriptional upstream repressor of a RAG program, suggesting that it could be a potential suppressor of regeneration. Conversely, since REST is a repressor of a pro-regenerative program, these analyses predict that counteracting REST would enhance regeneration after injury. To formally test this model, we next performed several experiments both in vitro and in vivo, using DRG neurons cultured on a growth-suppressive substrate to model the CNS-injured environment, along with two different in vivo models of CNS injury – complete spinal cord injury (SCI) and optic nerve crush.

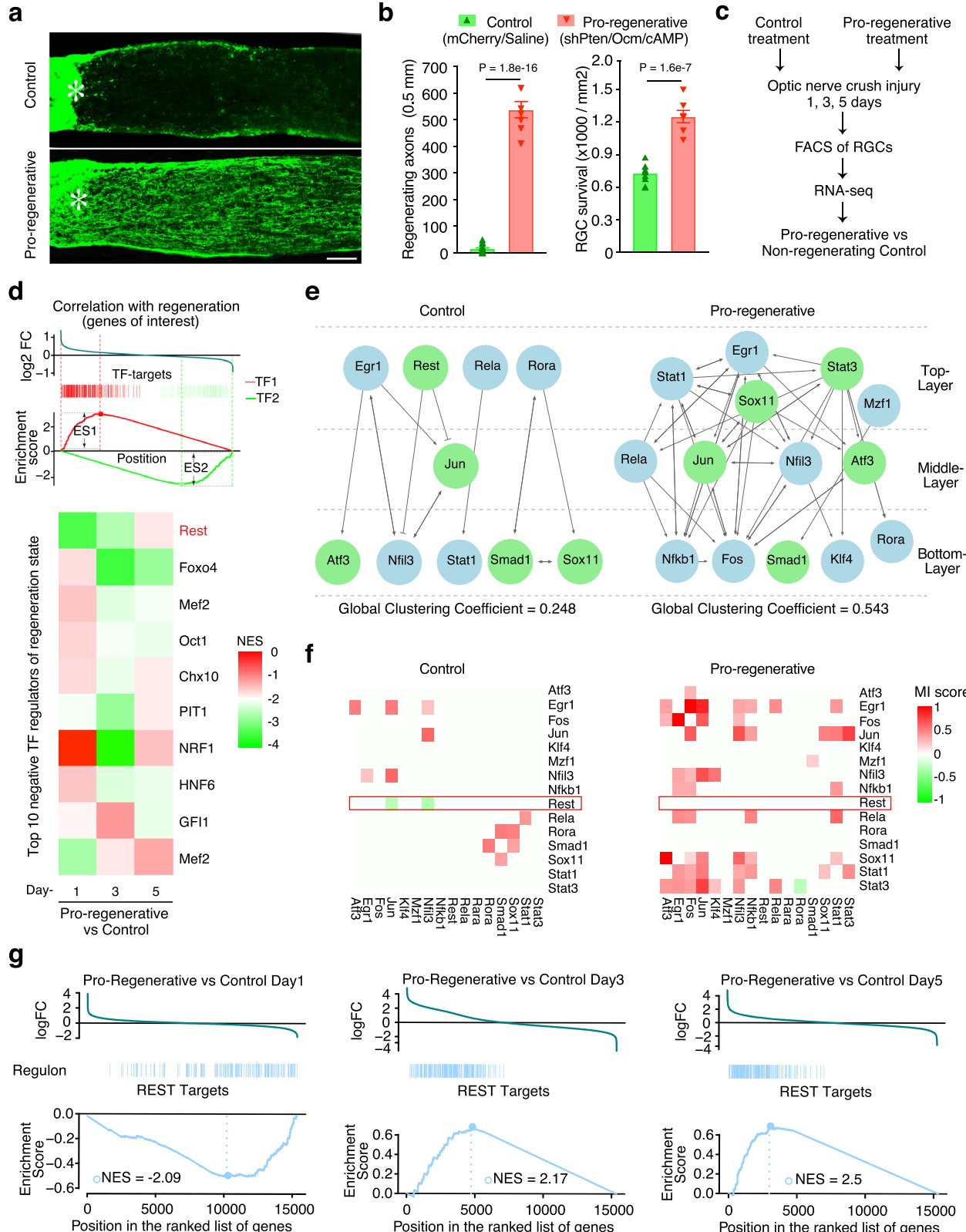

### REST deletion facilitates, and over-expression inhibits, neurite growth in vitro

We first tested the consequences of gain- and loss- of function of REST in dissociated adult DRG neurons in vitro. We hypothesized that if REST were indeed inhibitory, its depletion should be permissive, whereas its overexpression would inhibit the normal ability of PNS neurons to extend processes. REST depletion was achieved by infecting DRG neurons obtained from REST$^{flx/flx}$; STOP$^{flx/flx}$TdTomato mice (Methods) with adeno-associated virus expressing Cre recombinase (AAV-Cre; Methods). Cells infected with AAV-Cre, but not control virus AAV-GFP, showed reduced REST mRNA and protein levels with tdTomato expression turned on (Supplementary Fig. 7a–c).

To test the role of REST in a growth-suppressive environment that mimics the injured CNS, we grew DRG neurons on chondroitin

**Fig. 4 | REST is a transcriptional repressor negatively correlated with CNS regenerative state. a** Longitudinal sections through the mature mouse optic nerve immunostained for GAP43 two weeks after optic nerve crush. Control mice received an intraocular injection of AAV2-shLuciferase.mCherry 2 weeks before crush and saline immediately afterwards, while mice receiving pro-regenerative treatment were injected with AAV2-shPten.mCherry before crush, and oncomodulin (Ocm) and CPT-cAMP (a co-factor of Ocm) immediately after nerve injury. **b** Quantitation of axon growth (left) and RGC survival (right). Bars represent mean axon growth ± SEM. *Asterisk* in: nerve injury site. Scale bar in: 120 μm. Statistical significance was assessed with two-tailed *t*-test. *n* = 7 mice in the Control group and *n* = 10 mice in the Pro-regenerative group. **(c)** Overview of RNA-seq of FACS-purified RGCs receiving control or pro-regenerative treatments. n = 5 replicates in each condition. **d** Gene set enrichment analysis (GSEA) to screen TFs correlating with RGC regenerative state. Upper panel: schema demonstrating the principle of GSEA, which determines whether a TF's targets are randomly distributed, primarily found at the genes up-regulated by pro-regenerative treatments (logFC > 0) or at the genes down-regulated (logFC < 0). An enrichment at the bottom suggests that the TF down-regulates genes of interest, and is thus a negative regulator of the regenerative state (ES < 0; TF2 as an example), while an enrichment at the top suggests this TF is a positive regulator of regeneration (ES > 0; TF1 as an example). Bottom panel: A total of 1137 TF targeted gene sets (Methods) were screened and the top 10 negative TF regulators of RGCs' regeneration state were shown in the heatmap by their normalized enrichment scores (NES). **e** Transcriptional regulatory networks comparing RGCs in non-regenerating (control) and regenerating state (pro-regenerative). The networks were constructed using the TF-network pipeline described in Fig. 2a. **f** Mutual information (MI) scores of each TF-pair in the networks **e** indicating the degree of their correlation. **g** Distribution of REST-repressed target genes defined by ARACNe throughout the de-regulated genes by pro-regenerative treatments ranked by log2-fold changes (logFC, pro-regenerative vs non-regenerating) at indicated times following optic nerve crush.

sulphate proteoglycans (CSPG), a class of growth-suppressive extra-cellular matrix molecules present in injured CNS tissue[55], and compared this with growth on laminin, a growth-permissive molecule that positively supports extension of injured peripheral axons[56]. We first determined a CSPG dose that inhibits neurite growth without affecting cell survival (Supplementary Fig. 7d; Methods) and used this concentration to test the effects of REST depletion in DRG neurons. In agreement with previous findings[57], DRG neurons treated with AAV-GFP had limited neurite extension when cultured on CSPG (Fig. 6a, b). However, REST reduction induced by AAV-Cre (Fig. 6c) enhanced neurite outgrowth by ~40% compared with control neurons (Fig. 6a, b, CSPG group), showing that inhibition of REST enables neurite extension of regeneration-competent neurons in a growth-suppressive environment. REST deletion did not affect neurite extension of DRG neurons when cultured on laminin (Fig. 6a, b, laminin group), suggesting that REST-mediated inhibition of growth processes may be activated by a growth-suppressive environment that mimics the injured CNS, such as the presence of CSPG, but is not seen in the presence of permissive substrates that support peripheral axonal growth.

We further hypothesized that REST over-expression might inhibit the ability of DRG neurons to extend processes following a PNS injury. To test this hypothesis, we over-expressed REST in cultured DRG neurons for seven days using lentiviral constructs, followed by re-plating to remove existing neurites in vitro. This model recapitulates many biochemical and morphological features of an in vivo pre-conditioning peripheral nerve injury (Methods)[58]. The efficiency of REST over-expression was confirmed by qPCR (Supplementary Fig. 7e). Increasing REST concentration dose-dependently inhibited neurite extension, particularly at the highest concentration (Fig. 6d).

## REST deletion enhances corticospinal tract (CST) regeneration after spinal cord injury

To test the predicted role of REST in CST axon regeneration in vivo, we injected AAV-GFP or AAV-Cre into the sensorimotor cortex of adult REST^flx/flx mice[59], where CST neuronal cell bodies are located. Following sham or T10 SCI, CST axons were traced by injecting the anterograde tracer biotinylated dextran amine (BDA) into the sensorimotor cortex (Fig. 7a). No difference was found in astrogliosis and lesion size between mice receiving AAV-GFP or AAV-Cre (Supplementary Fig. 8a–c). At 8 weeks post injury, CST axons in mice receiving AAV-GFP exhibited characteristic dieback from the lesion center, consistent with previous reports[60]. Conditional deletion of REST led to ~45% more CST axons proximal to the lesion site (Fig. 7b, c; Supplementary Fig. 8d), suggesting either a lack of dieback in the axons of REST-deficient neurons or a regrowth of axons after injury.

To distinguish between these potential mechanisms, we first examined CST axons 3 days post-injury. Apparent dieback and large numbers of retraction bulbs were observed at this early time point in both control and REST-deleted axons (Supplementary Fig. 9a). We then measured branching of CST axons at 4 weeks post injury which, when increased, is considered to be strong evidence of regenerative growth[41] (Fig. 7d; Methods). Mice receiving AAV-Cre displayed far more branching from injured CST axons in the area proximal to the lesion center than controls (Fig. 7e, f), indicating that REST depletion promotes regenerative axon growth. In addition, REST-deficient CST axons traced by BDA expressed more GAP43 (Fig. 7g, h, GAP43 + BDA + ) and synaptophysin (Fig. 7i, j, Syn+ BDA + ) than wild-type axons, especially in bouton-like structures in gray matter just proximal to the lesion, indicating the potential of these axons to re-grow and potentially establish pre-synaptic machinery. REST deletion in uninjured mice did not change the number of CST axons (Supplementary Fig. 9b), suggesting that the lack of REST does not affect axon growth in intact or homeostatic states.

## REST inactivation stimulates optic nerve regeneration and RGC neuroprotection

We next tested the role of REST in optic nerve regeneration by intraocular injection of an RGC-selective adeno-associated virus expressing a previously validated dominant-negative REST mutant (AAV2-d/n REST) that includes the DNA-binding domain but lacks the repressor domain of REST[61] vs. a control virus (AAV2-GFP: Supplementary Fig. 10a; Methods). After allowing one week for expression of virally encoded d/n REST, we dissected and dissociated retinas and placed the cells in culture[62] with or without recombinant oncomodulin, forskolin (to elevate cAMP), and mannose, a necessary co-factor[47]. Expression of d/n REST caused a modest increase in neurite outgrowth by itself and greatly enhanced levels of neurite outgrowth induced by Ocm/cAMP/mannose (Fig. 8a, b). In addition, d/n REST increased RGC survival irrespective of the presence or absence of Ocm/cAMP/mannose (Fig. 8a, c).

To validate these observations in vivo, we used two independent methods to counteract REST (Supplementary Fig. 10a). In the first approach, we examined whether AAV2-d/n REST would induce optic nerve regeneration and/or promote RGC survival. Two weeks after optic nerve injury, expression of d/n REST was sufficient to stimulate 43% of the level of axon regeneration (Fig. 8d, e) that was achieved with the powerful combinatorial treatment (*pten* deletion, rOcm, CPT-cAMP) that we used to generate the transcriptome dataset (c.f. Figure 4a, b). In addition, d/n REST expression more than doubled RGC survival at two weeks post-optic nerve injury (compared to mice injected with AAV2-GFP: Fig. 8f), an effect that fully recapitulated the strong neuroprotection afforded by the combination of *pten* deletion, rOcm, and CPT-cAMP (Fig. 4b). In parallel to our cell culture studies (Fig. 8a–c), we also examined the effect of combining d/n REST expression with Ocm plus cAMP in vivo. Whereas a single injection of rOcm + cAMP alone induced little regeneration and no increase in RGC survival relative to untreated controls, combining rOcm + cAMP with

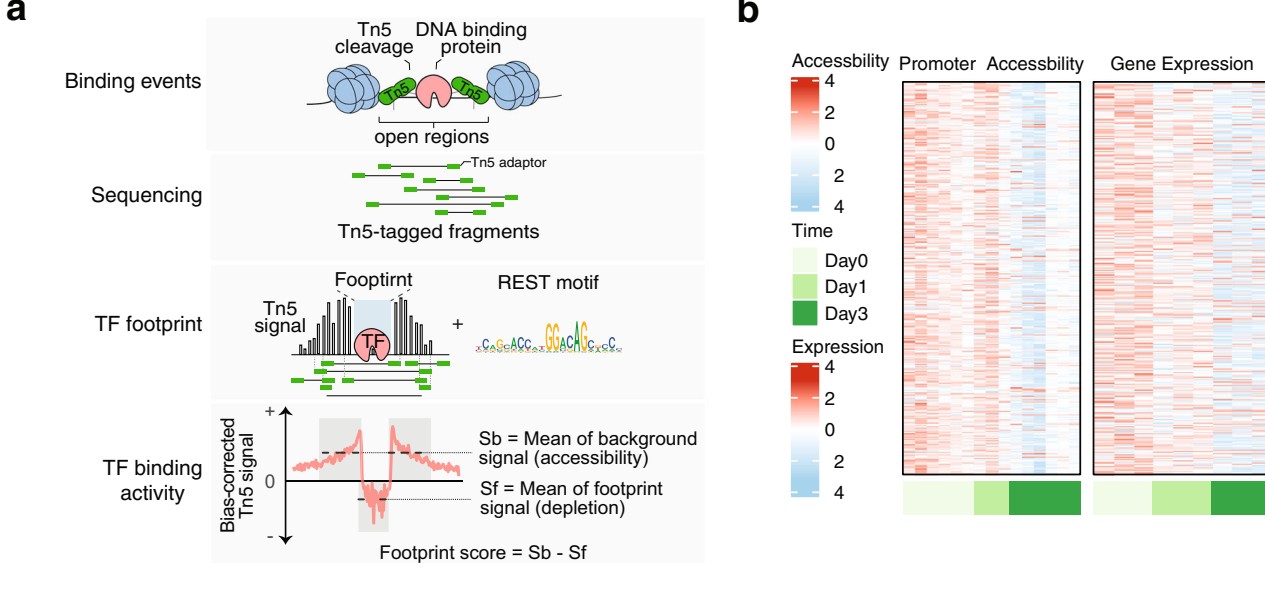

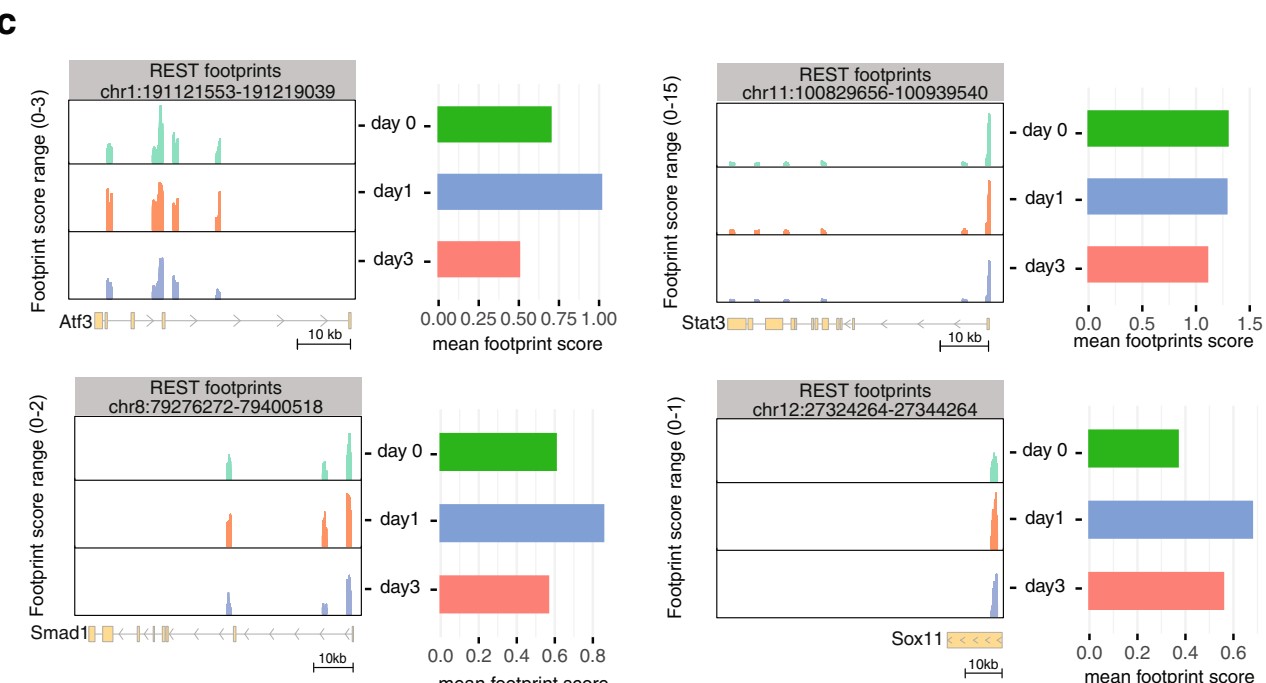

**Fig. 5 | REST foot-printing in CNS-injured neurons. a** Schematic diagram depicting analysis to identify REST binding sites using DNA footprinting analysis of ATAC-seq data generated from RGCs FACS-purified at 0 (sham) 1, 3 days following optic nerve crush[54]. During ATAC, Tn5 transposase cleaves DNA free of chromatin-bound proteins such as transcription factors (yellow) and inserts sequencing adapters (green). Tn5-tagged DNA fragments are sequenced to yield reads, and then mapped to the genome to create signals of single Tn5 insertion events (black bars), in which TF binding is visible as depletion of signals (defined as footprint). DNA footprints overlapping with REST motifs are defined as direct REST binding sites. REST binding activities can be quantitated by scoring each footprint's depleted signal and the surrounding chromatin accessibility, correlating with the presence of a TF at its target loci, and the chromatin accessibility of the regions where this TF binds. **b** Promoter (±2 kb of a gene's transcription start site) accessibility and expression changes of 801 REST-targeted genes at indicated time points following CNS injury. Normalized ATAC-seq and RNA-seq counts scaled by row are displayed in the heatmaps. **c** Genome-browser views of REST footprinted sites and barplots of mean REST footprint scores within indicated genomic distances of the regenerative TFs (*Atf3, Stat3, Smad1, Sox11*). The footprint scores are calculated with TOBIAS as described in **a**, indicating REST binding activities at these genes. The genomic distance covers the entire gene from 2 kb upstream of a gene's transcription start site, to 1 kb downstream of the end of the gene. The coding exons of a gene are displayed as yellow boxes connected by horizontal lines representing introns. Arrowheads on the connecting intron lines or the coding exons indicate the direction of transcription.

expression of d/n REST increased axon regeneration 55% above the level achieved with d/n REST expression alone (Fig. 8d, e). RGC survival was elevated to the same extent as with d/n REST expression alone (Fig. 8f).

As an alternative approach (Supplementary Fig. 10a) to investigate the role of REST in vivo, we deleted it in mature RGCs via AAV2-Cre-driven recombination in mice with homozygous conditional REST alleles and the same TdTomato reporter line (REST^flx/flx; STOP^flx/flx

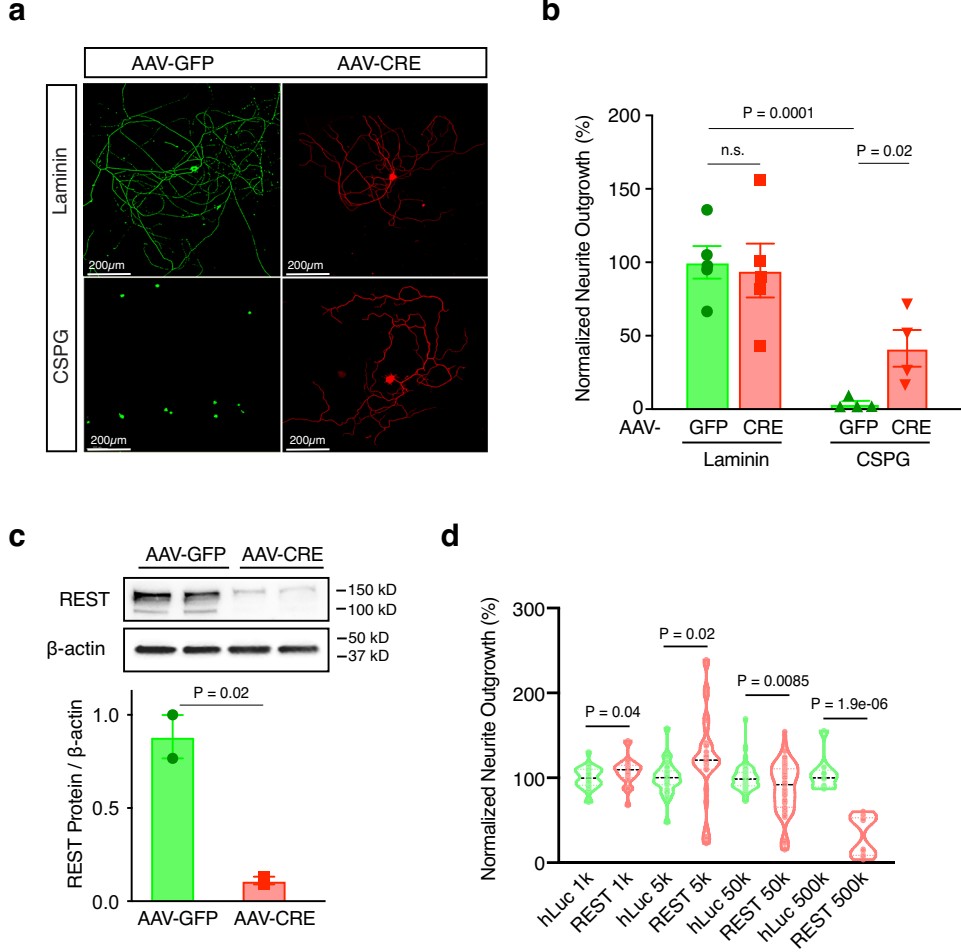

**Fig. 6 | REST inhibits neurite growth in vitro. a** Tuj1 (βIII tubulin) staining of REST $^{flx/flx}$;tdTomato DRG neurons cultured on CSPG (5 µg/ml) or laminin only (2 µg/ml) and transduced with AAV-GFP (green) or AAV-CRE (red) at ~100,000 genome copies per cell for 7 days to allow the expression of transgenes. **b** Quantitation of neurite outgrowth normalized to AAV-GFP infected neurons cultured on laminin. Data are presented as mean neurite outgrowth ± SEM. $N = 3$ replicate wells in each condition examined over at least 3 independent experiments. **c** Representative western blot and quantitation of REST levels in DRG cells transduced with AAV-GFP or AAV-CRE. Data are presented as mean ± SEM. $N = 3$

replicate wells in each condition examined over two independent experiments. **d** Volcano plot showing the mean neurite outgrowth of re-plated DRG neurons infected with lentiviral constructs expressing either REST (Lv135-REST) or humanized luciferase protein (Lv135-hLuc) as a control driven by the CMV promoter at indicated genome copies per cell for 7 days. Neurite extension was quantified 24 h following re-plating. Data are presented as mean neurite outgrowth ± SEM normalized to control at indicated viral doses. $N = 6$ replicate wells in each condition examined over at least 3 independent experiments. For **b**–**d**, statistical significance was assessed by two-tailed t-test for indicated comparisons.

TdTomato) used in the CST repair studies (see Methods). AAV2-Cre was injected into one eye of REST$^{flx/flx}$; STOP$^{flx/flx}$TdTomato mice, while the contralateral control eye received an injection of AAV2 expressing GFP. REST knock-out was confirmed by tdTomato expression in the retinas of REST$^{flx/flx}$; STOP$^{flx/flx}$TdTomato mice exposed to AAV2-Cre, whereas no TdTomato was observed in control retinas receiving AAV2-GFP. Conditional deletion of REST in RGCs, similar to expression of d/nREST, induced considerable axon regeneration (Fig. 8d, e), in this case averaging ~ 50% of the level induced by the three-way combination of *pten* deletion, rOcm, and CPT-cAMP (Fig. 4b). Negative controls were pooled for the different genotypes and viruses used in these studies based on the lack of significant differences in outcomes among controls for AAV2-Cre plus REST$^{fl/fl}$ (strain C57/B6, Mean ± SEM: 71.07 ± 14.65) and for AAV2-d/nREST injections in wild-type 129S1 mice (Mean ± SEM: 41.57 ± 13.65; $P = 0.09$; see legend for Fig. 8). In addition, as observed with d/n REST expression, deletion of REST in RGCs doubled the level of RGC survival above that seen in control retinas two weeks after optic nerve injury (Fig. 8f), an effect similar to that achieved with the combinatorial treatment used to generate the transcriptional dataset.

Deletion of *pten* is perhaps the most effective single treatment for inducing optic nerve regeneration described to date[4]. On average, counteracting REST captured ~ 2/3 of the effect of *pten* deletion on axon regeneration (Fig. 8e) and the full effect of *pten* deletion on RGC survival (Fig. 8f). Thus, REST can be considered a major suppressor of RGC survival and optic nerve regeneration in mature mice. We also investigated whether *pten* deletion would occlude the effects of counteracting REST, which would suggest that the two share common effector pathways, or whether they might instead show some degree of additivity. Our results point to partially additive effects on axon regeneration (Fig. 8e), suggesting at least some independence of effector pathways.

Accompanying its effects on RGC survival and axon regeneration, expression of d/n REST increased expression of several regenerative TFs (ATF3, SOX11, pSTAT3, pCREB) in the TF regulatory network in RGCs, as assessed by immunostaining retinal sections 1 day after optic nerve injury (Fig. 8g, h). At day 7, expression of genes associated with regeneration and/or survival, including *Sprr1a*, *Bdnf* and *Gap-43*, were found to be increased based on qPCR using mRNA from FACS-sorted RGCs 7 days after optic nerve injury (Fig. 8i: *$P < 0.05$, **$P < 0.01$;

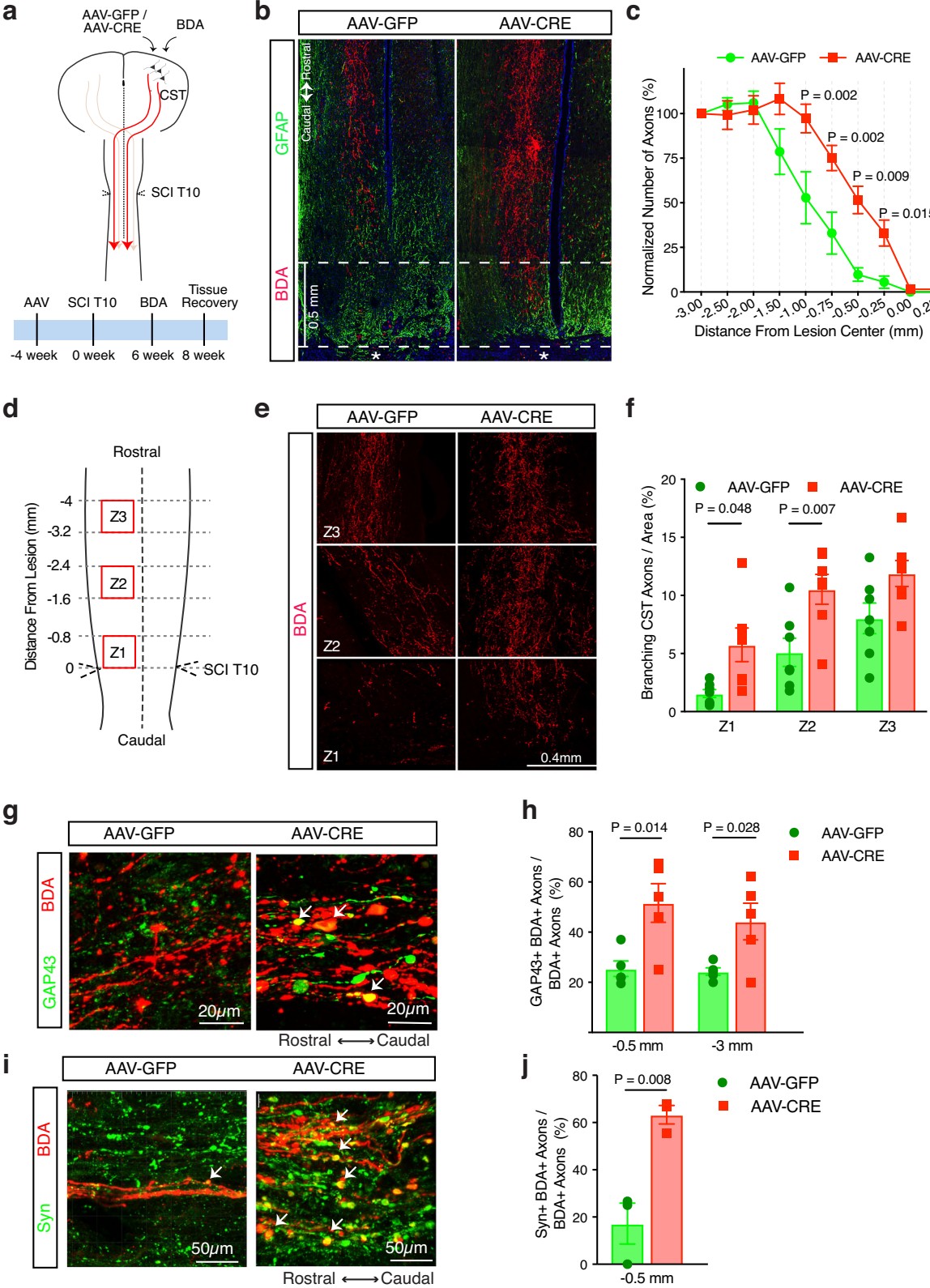

Methods). These findings are consistent with the elevated expression of key regenerative TFs and effector genes associated with axon growth that we observed in REST-depleted cortical motor neurons, and show that, as with spinal cord injury, REST antagonism enhances central axon regeneration. Thus, we were able to confirm the predicted repressive effects of REST on regeneration based on our systems genomic analysis in two distinct models of CNS injury.

## Discussion

The present study used a stepwise, systems genomics approach to predict upstream transcriptional regulators of intrinsic regeneration-associated gene expression programs in the nervous system. Multiple independent bio-informatic analyses were used to evaluate existing and newly produced gene expression datasets, all of which converged on the hypothesis that the transcriptional repressor, REST, is a

**Fig. 7 | REST deletion enhances corticospinal (CST) axon regeneration after anatomically complete spinal cord crush injury. a** Schematic diagram and timeline of inducing REST deletion and SCI lesions. **b** Confocal images of BDA-labeled CST axons of lesioned spinal cord also stained for astrocytes (glial fibrillary acidic protein, GFAP). Dashed line represents lesion center (marked with *). **c** Intercepts of CST axons with lines drawn at various distances rostral to the lesion center were counted and expressed as percent of the number of intact axons at 3 mm proximally to control for potential variability in the fluorescence intensity among animals. Each dot represents mean ± SEM; $n = 10$ mice in AAV-GFP group and $n = 12$ in AAV-CRE group. Statistical significance was assessed by two-way ANOVA with repeated measures and Bonferroni post-hoc test, comparing AAV-CRE to AAV-GFP at each distance. **d** Schematic diagram showing regions along the central canal in horizontal sections of lesioned spinal cord used for quantifying branching of CST axons. **e** Confocal images of CST axons labeled by BDA in Z1, Z2, and Z3, three $0.8 \times 0.8$ mm$^2$ squares drawn in the gray matter of each spinal cord. **f** Quantitation of the number of axons per area. **g–j** The number of GAP43- or Synaptophysin- expressing axons co-labeled with BDA were counted at 0.5 mm or 3 mm rostral to the SCI crush, and are expressed as percent of BDA labeled axons at respective distances. Confocal images of CST axons (BDA) co-labeled with **g** GAP43 or **i** Synaptophysin (Syn) at 0.5 mm rostral to the lesion center. **h** Quantitation of CST axons expressing GAP43 at 0.5 and 3 mm rostral to lesion center. **j** Quantitation of CST axon terminals expressing Syn at 0.5 mm rostral to lesion center. All bars represent mean ± SEM. Statistical test: **f** two-way ANOVA with Bonferroni post-hoc test, $n = 7$ mice per condition; **h, j** two-tailed $t$-test compared to AAV-GFP in each area. $n = 5$ mice per condition in **h** and 3 mice per condition in **j**.

potential upstream negative regulator of a regenerative gene expression program in the CNS (Fig. 1a). We then experimentally demonstrated that disrupting REST activates a core molecular program driven by a TF network similar to the one activated during peripheral nerve regeneration (Fig. 1b). These findings in turn predicted that counteracting REST would substantially improve regeneration, which we then confirmed using two well established models of CNS injury, the optic nerve and the corticospinal tract (CST) (Fig. 1c). Together, our data are consistent with a model whereby REST acts by suppressing the interaction and the expression of pro-regenerative TFs within the RAG network and by directly regulating expression of many effector genes, consistent with its known function as a transcriptional suppressor. Indeed, direct REST binding to genes encoding multiple regeneration-associated TFs and effector genes was confirmed by DNA footprinting analysis. These results demonstrate that REST represses CNS regeneration in vivo, and conversely, that its depletion or inhibition by expressing a dominant-negative mutant enhances CNS regeneration.

Our first step was to use an unsupervised, step-wise bio-informatic approach to characterize the regulatory network structure of regeneration-associated TFs (Fig. 2a). From this analysis, we identified a core set of five TFs (Jun, SMAD1, Sox11, STAT3 and ATF3) forming regulatory network that was conserved across all PNS datasets (Fig. 2c). Each of these core pro-regenerative TFs is increased early after PNS injury (Supplementary Fig. 1a), in agreement with previous findings of their essential role during PNS regeneration[8,36,38,39,63,64], and each connection of TF pairs is experimentally supported[33,34], adding confidence to our bio-informatic predictions. By contrast, in the non-regenerating CNS (spinal cord and optic nerve), this network loses its tiered structure and instead adopts a simpler, less inter-connected, dismantled structure (spinal cord: Fig. 2d; optic nerve: Fig. 4e, f). Remarkably, CNS neurons with enhanced regenerative capacity induced by combined genetic and molecular manipulations regain the complex, multi-layer TF network with higher inter-connectivity (Fig. 4e, f), similar to the TF network induced in the regenerating PNS (Fig. 2c). In the dismantled CNS network, REST appears as a top-tier negative regulator, predicted to inhibit other lower-level TFs. The prediction of REST being a transcriptional repressor was further supported by an independent TF-screening approach that evaluated ~1000 TFs and their experimentally-proven target genes, identifying REST as a top negative regulator of the gene set activated in regenerating CNS neurons (Fig. 4d). One limitation of the TF-network or GSEA-based approaches is the reliance on non-neuronal ChIP-seq datasets. For this reason, we provided extensive evidence from neurons in the context of CNS injury to support our bio-informatic predictions about REST targets.

Our hypothesis was supported by the findings that *Rest* was specifically upregulated across multiple CNS injury datasets (Supplementary Fig. 1a, Fig. 3b), and that *REST* displays repressor-like activity on chromatin, decreasing promoter accessibility and the expression of RAGs predicted to be bound by REST (Fig. 5). When REST is inhibited,

Jun, STAT3, Sox11 and ATF3, all members of the core TF regulatory network, are up-regulated both in injured cortical neurons (Fig. 3b) and in RGCs (Fig. 8g–i). Importantly, each of these TFs has been shown independently to promote axonal regeneration, including in the injured CNS in some cases[9,10,63,65]. These observations, coupled with our data, supports a model whereby the up-regulation of the core TFs following REST deletion directly contributes to regenerative growth, which was further validated in multiple experimental models of CNS injury.

We also note that the effect of counteracting REST was considerable: regeneration induced by counteracting REST was approximately 2/3 of that induced by PTEN deletion, a treatment that provides perhaps the strongest regeneration induced by a single genetic manipulation to date[4], and roughly half the robust level of axon regeneration induced by *Pten* deletion combined with Ocm and cAMP elevation (Fig. 8d, e), the potent combinatorial treatment used to generate our original regeneration RNA Seq dataset. Combining d/n REST expression with Ocm plus CPT-cAMP brought the level of regeneration even closer to that induced by the strong combinatorial treatment, while a combinatorial treatment to knock down PTEN and counteract REST in RGCs led to considerably greater regeneration than either one alone. REST depletion was sufficient to double levels of RGC survival, affording the same level of neuroprotection as either combinatorial therapy or PTEN deletion alone, which is notable, since to date, few factors other than PTEN deletion enhance both RGC regeneration and survival. For example, although ATF3 is pro-survival, it has no effect on RGC regeneration[66]; Sox11 is pro-regenerative, but when overexpressed, leads to the death of alpha RGCs[10]; and STAT3 is pro-regenerative, but does not increase survival[67].

Although REST is among the most widely studied transcription factors in the CNS[24–26], there was no previous evidence to link REST specifically to CNS repair. One recent study examined REST in the context of peripheral injury in vitro, primarily in embryonic sensory neurons, focusing on its upstream epigenetic regulators. That study showed that REST expression transiently increases in response to peripheral injury, but is quickly repressed by an epigenetic regulator, UHRF1, which interacts with DNA methylation enzymes to restrict the transcription of REST, as well as PTEN, a suppressor of cell-intrinsic growth[68]. We did not observe significant changes of *Rest* expression levels across multiple PNS injury models at different time scales (Supplementary Fig. 1a, PNS1-5). These findings suggest that the expression levels of REST or other intrinsic growth suppressors are tightly controlled in peripherally injured neurons to allow peripheral nerve regeneration[69].

How REST interacts with PTEN and other pro-regenerative manipulations requires further investigation. As a protein phosphatase, PTEN antagonizes the PI3K-AKT-mTOR pathway to inhibit protein translation, cell cycle progression and cell survival[64], as well as transcriptional regulation of cell-growth-associated genes[70,71]. Our findings indicate that REST is likely not acting by eliciting changes in PTEN or the downstream canonical mTOR pathway to regulate regeneration, as

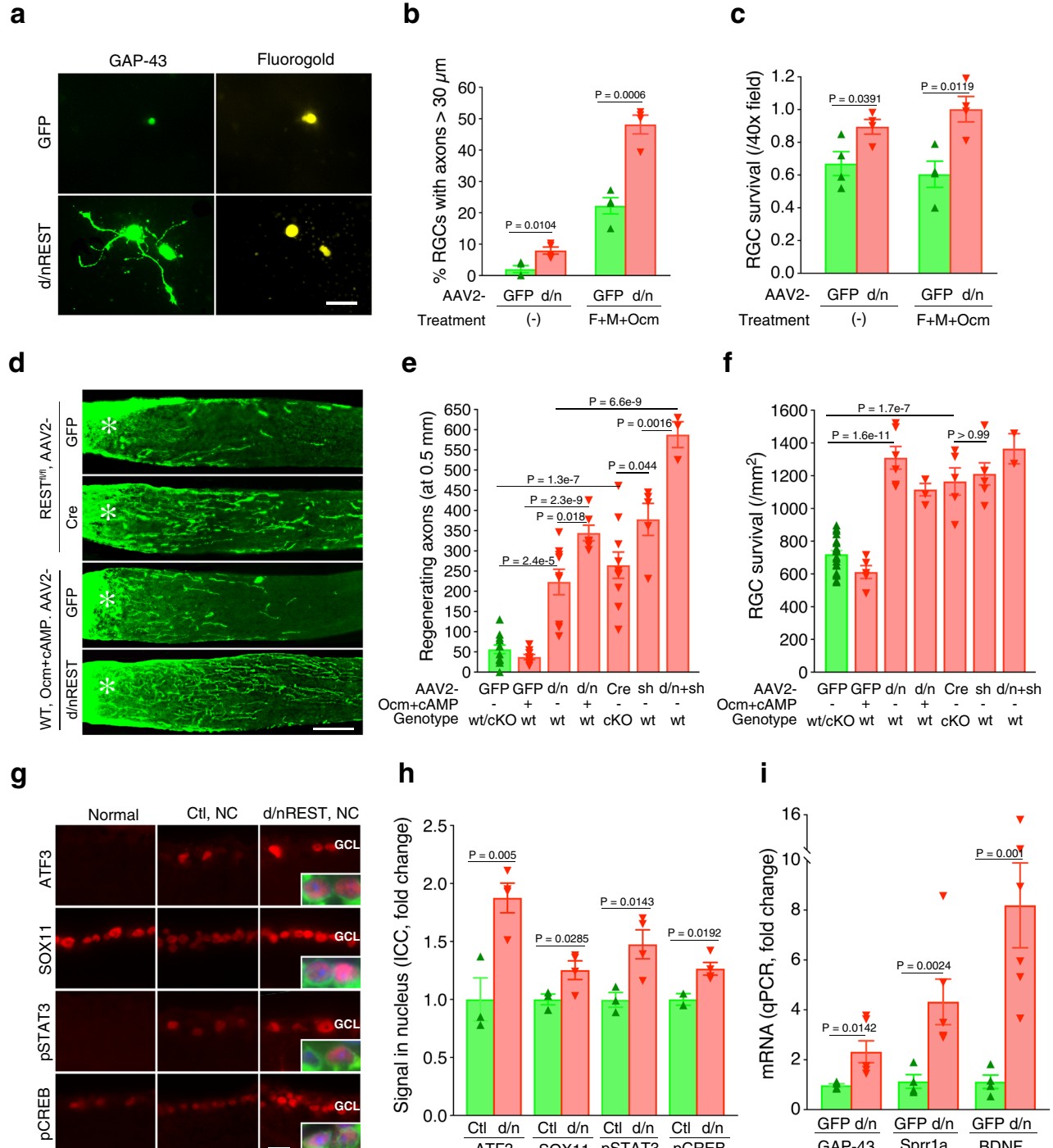

**Fig. 8 | REST inactivation stimulates axon outgrowth from RGCs, optic nerve regeneration, and RGC neuroprotection. a–c** Effect of REST inactivation on adult rat RGCs in culture. Mice received intraocular injections of either AAV2-d/nREST (d/nREST) or AAV2-GFP (GFP) one week prior to retina dissociation. RGC culture was maintained in the presence or absence of forskolin (to elevate cAMP), mannose, and recombinant oncomodulin (F + M + Ocm) for 3 days. **a** GAP-43 immunostaining of RGCs (identified via retrograde labeling with Fluorogold injected into the superior colliculus 7 days earlier). **b** Axon outgrowth represented as percentage of RGCs with axons ≥ 30 μm. **c** RGC survival in culture. **d–f** Effects of REST deletion or antagonism on optic nerve regeneration and RGC survival in vivo. REST deletion was obtained by intraocular injection of AAV2-Cre in REST^flx/flx mice. REST antagonism was obtained by intraocular injection of AAV2-d/nREST (d/n) in wildtype (WT) mice. In addition to inactivating REST, some WT mice received recombinant Ocm plus CPT-cAMP (Ocm + cAMP). Control mice (green bar in E, F) were pooled from REST^flx/flx mice and WT receiving AAV-GFP. **d** Longitudinal sections of

CTB-labeled axons through the optic nerve. *Asterisk*: nerve injury site. **e** Quantitation of regenerating axons 500 μm distal to the injury site and **f** RGC survival. **g, h** ATF3, SOX11, pSTAT3, and pCREB changes one day after nerve crush in RGC receiving control (Ctr, AAV2-GFP) or AAV2-d/n-REST (d/n). **g** Representative immunohistochemistry images of RGCs. Inserts show higher magnification. TUJ1: RGCs, *green*; DAPI: nuclei, *blue*; target genes: *red*. **h** Quantitation of RGCs expressing each REST target genes. **i** *Gap43*, *Sprr1a* and *Bdnf* mRNA levels seven days after nerve crush in FACS-selected RGCs expressing GFP or d/n REST. All bars represent mean ± SEM. Statistical tests: **b**, **c**: two-tailed *t*-test, *n* = 4 biological replicates in each condition; **e**, **f**: one-way ANOVA with Bonferroni post-hoc test, *n* = 12 mice for GFP/-, *n* = 8 mice for GFP/+, *n* = 9 mice for d/n/-, *n* = 6 mice for d/n/+, *n* = 10 mice for Cre/-, *n* = 5 mice for sh/-, *n* = 3 mice for d/n+sh/-; **h, i**: multiple two-tailed *t*-test, *n* = 4 mice for Ctr and *n* = 6 mice for d/n. Scale bar in a: 20 μm, in D: 200 μm, in g: 15 μm.

our gene expression data show no change in the levels of *Pten* with REST deletion (Supplementary Fig. 4e), nor do we see changes in phosphorylation of ribosomal protein S6, which would be indicative of changes in the mTOR pathway (Supplementary Fig. 10b, c). In addition, we observed additive effects of *Pten* deletion combined with counteracting REST, suggesting that the two treatments may activate downstream effector pathways that are at least partially separate (Fig. 8e–h).

Further studies will also be required to clarify the precise molecular mechanisms by which REST itself is regulated in the context of PNS or CNS injury. Others have shown that REST can be regulated post-transcriptionally[72], post-translationally via ubiquitination/ deubiquitination[73], and by cytoplasmic sequestration[74]. Thus, investigating how REST is regulated in CNS neurons in growth-permissive or non-permissive states may further illuminate non-transcriptional mechanisms underlying PNS or CNS regeneration. We also note that growth across the lesion boundary in a complete spinal cord injury or in other forms of CNS injury will likely require both intrinsic growth cues and external growth facilitators to activate regeneration-associated genes and pathways in CNS neurons, with an appropriate lesion-bridging substrate[75]. Lastly, it is also plausible that neuronal regulation by REST as demonstrated in the current study is one of several mechanisms by which its deletion promotes cell-intrinsic growth. REST is expressed in non-neuronal cells in the CNS including astrocytes, regulating genes involved in inflammatory processes[76], which might affect neuronal reponses to injury and axon regeneration. Further work including generating astrocyte-specific REST depletion will be required to more fully understand non-neuronal-associated mechanisms.

## Methods

All animal testing and research was carried out in compliance with ethical regulations laid out by the National Institutes of Health (NIH) guide for the care and use of animals. All experiments were performed in compliance with protocols approved by the Institutional Animal Care and Use Committee at University of California Los Angeles and Boston Children's Hospital.

### Animals

Mouse lines, including 129S1, C57BL/6 J, loxP-REST-loxP (REST^flx/flx), B6.Cg-Tg(Thy1-CFP)23Jrs/J, and Rosa26-CAG-loxP-STOP-loxP-tdTomato (STOP^flx/flx TdTomato), were purchased from Jackson Laboratory. REST^flx/flx; tdTomato homozygous mice were generated by crossing REST^flx/flx[59] and STOP^flx/flx TdTomato mice. Young adult mice between 4 and 6 weeks old including both sexes were used for all experiments in spinal cord studies and 8–12 week old animals in optic nerve regeneration studies.

### Spinal cord injury and corticospinal tract (CST) injections

Surgical procedures for spinal cord injury and CST injections in mice were similar to those described previously[60,75,77], and were conducted under general anesthesia with isoflurane using an operating microscope (Zeiss, Oberkochen, Germany), and rodent stereotaxic apparatus (David Kopf, Tujunga, CA). The adeno-associated virus-green fluorescent protein (AAV-GFP) or adeno-associated virus expressing Cre recombinase (AAV-Cre) were obtained from Boston Children's Hospital Viral Vector Core. The viruses referred to as AAV-GFP and AAV-Cre were AAV2/8.CAG.eGFP.WPRE.polyA and AAV2/8.CAG.Cre-HA.WPRE.polyA, respectively. A total of 2 μl AAV2/8-GFP or AAV2/8-Cre virus at a titer of ~10^13 gc/ml was injected into the left cerebral motor cortex at the following coordinates (in mm): anteroposterior/mediolateral: 0.5/1.5, 0.0/1.5, −0.5/1.5, −1.0/1.5, at a depth of 0.5 mm. Four weeks later, a laminectomy was performed at T10, and the spinal cord was crushed using .1mm-wide customized forceps. To trace corticospinal tract axons, 2 μl biotinylated dextran amine 10,000 (BDA,

Invitrogen, 10% wt/vol in sterile saline) was injected at the same coordinates as the AAVs into the left motor cortex six weeks after SCI. Mice that underwent surgical procedures were placed on a warming blanket and received an analgesic before wound closure and every 12 h for 48 h post-injury.

### Immunostaining of spinal cord and cortex

Spinal cords were recovered and stained as previously described[60,75]. Following terminal anesthesia by pentobarbital, mice were perfused transcardially with 10% formalin (Sigma). Spinal cords and brains were removed, post-fixed overnight, transferred to buffered 30% sucrose for 48 h, embedded in O.C.T. Compound (Tissue-Tek, Sakura-Finetek/VWR) and cryostat-sectioned at 30 μm. Serial horizontal sections of spinal cord containing the lesion sites and brain containing the viral injection sites were cut and processed for immunostaining. The following primary antibodies were used: anti-GFAP (DAKO Z0334, 1:1000, free-floating), anti-GAP43 (1:1000, Benowitz lab), anti-Synaptophysin (Synaptic Systems 101004, 1:1000, free-floating), and RFP (1:500, Abcam ab62341, free-floating). BDA tracing was visualized with streptavidin-HRP (1:300, PerkinElmer SAT704A001EA) antibodies plus Cy3-TSA (1:200, PerkinElmer SAT704A001EA). Sections were cover-slipped using Prolong Diamond Antifade Mounting media with DAPI (ThermoFisher) to stain cell nuclei. Each section was imaged on ZEISS LSM 880.

### Quantitation involving CST axons

To quantify total labeled CST axons, we counted intercepts of BDA-labeled fibers with dorsal-ventral lines drawn at defined distances rostral to the lesion center. Similar lines were drawn and axons were counted in the intact axon tract 3 mm proximal to control for potential variability in the fluorescence intensity among animals. Fibers were counted on at least two sections per mouse, and the number of intercepts near or in the lesion was expressed as percent of axons in the intact tract divided by the number of evaluated sections. To quantify the number of branching axons from the main CST, three 0.8 × 0.8 mm² squares (Z1, Z2, Z3) were drawn along the central canal at defined distances rostral to the crush site. The number of axons were counted in each square, and are expressed as percent of area per section for each mouse. The number of GAP43- or Synaptophysin- expressing axons co-labeled with BDA were counted at 0.5 mm and 3 mm rostral to the SCI crush, and are expressed as percent of BDA labeled axons at respective distances. We examined BDA labeling 3 mm caudal to the lesion center to make sure the SCI lesions were complete. All axon counts were carried out by an investigator blind to the identity of the cases.

### Optic nerve crush and intraocular injections

Surgical procedures for optic nerve injury and intraocular injections in mice were similar to those described previously[5,46,62]. To investigate REST functions in vivo, we either deleted REST in RGCs or expressed a dominant-negative mutant form of REST[61] (d/n REST, gift of Dr. Gail Mandel, OHSU). For the former, REST^flx/flx-tdTomato mice received an intraocular injection of either AAV2-CAG-Cre.WPREpA (AAV2-Cre, to preferentially delete the gene in RGCs) or, as a control, AAV2-CAG-eGFP.WPREpA (AAV2-GFP). In the latter studies, 129S1 wildtype mice received AAV2-CAG-d/n human REST-HA-SV40pA (AAV2-d/nREST) to inactivate REST function or AAV2-GFP as a control. All viruses were injected in a volume of 3 μl and a titer of $1 \times 10^{13}$ gc/ml 2 weeks prior to optic nerve crush to insure adequate time for gene deletion or transgene expression at the time of nerve damage. Two days prior to the end of a 14-day survival period, cholera toxin B subunit (CTB, 3 μl/eye, 2 μg/μl, List Biological Laboratories, Inc., 103B) was injected intraocularly as an anterograde tracer to label axons regenerating through the optic nerve.

In some studies, 129S1 mice received an intraocular injection of AAV2-d/nREST or an AAV2 control virus two weeks before the optic

nerve crush and were euthanized at day 1 or day 7 after nerve injury. Retinas from these mice were prepared for immunostaining of serial sections (details in Methods: *Immunostaining of retinal sections and intensity quantitation*).

To investigate the transcriptome of RGCs during optic nerve regeneration or after counteracting REST, we carried out optic nerve crush surgery with different intraocular treatments in vivo, then used FACS to isolate RGCs for subsequent analyses (details in Methods: *FACS isolation of retinal ganglion cells*)

### Quantitation of optic nerve regeneration and RGC survival

Following transcardial perfusion with saline followed by 4% paraformaldehyde (PFA), optic nerves and retinas were dissected out and post-fixed with 4% PFA for 2 hours (RT). Nerves were transferred to 30% sucrose at 4 °C overnight before being frozen in O.C.T. Compound (Tissue-Tek, Sakura-Finetek/VWR) and sectioned longitudinally on a cryostat at 14 μm thickness. Regenerating axons were visualized by immunostaining for CTB (1:500, Genway Biotech, GWB-7B96E4) and were quantified in 4-8 sections per case to obtain estimates of the total number of regenerating axons at 0.5 mm distally from the injury site as described[46]. Whole retinas were immunostained for βIII-tubulin (1:500, free-floating. Abcam) to identify RGCs, and RGC survival was evaluated in 8 pre-designated fields in each retina as described[46].

### Immunostaining of retinal sections and quantitation of signals

Animals injected intraocularly with AAV2-d/nREST or a control virus underwent optic nerve crush surgery 14 days later and were euthanized and perfused after another 1 day or 7 days (Methods: *Optic nerve crush and intraocular injections*). Eyes were dissected out, post-fixed for 2 hours at RT, then transferred in 30% sucrose at 4 °C overnight. After embedding in O.C.T. and cryostat-sectioned at 14 μm, retinal sections were immunostained with primary antibodies against various proteins, including several transcription factors (anti-ATF3, 1:100, Abcam Ab207434; anti-SOX11, 1:500, Millipore ABN105; anti-pSTAT3, 1:200, Cell Signaling 9145; anti-pCREB, 1:100, Cell Signaling 9198; and anti-βIII tubulin [TUJ1], 1:500, Biolegend 801201 at 4 °C overnight followed by the appropriate fluorescent secondary antibodies the next day. Stained retinal sections were imaged using equal exposure conditions across all sections in both control and treated groups. Staining intensity was measured with Image J software on each individual RGC that was labeled by the TUJ1 antibody, and data were averaged from 50 − 100 consecutively encountered RGCs across 3 different areas from each retina, 3 − 4 retinas per group, and was compared between the control and treatment groups for each antibody.

### Retrograde labeling of RGCs and preparation of dissociated retinal cultures

The procedure for retrograde labeling of RGCs has been described previously[47,62]. Briefly, to distinguish RGCs from other cells in dissociated mixed retinal cultures, we injected 2% of Fluorogold (FG, Fluorochrome) into the superior colliculus (SC) bilaterally in adult rats. At the same time, rats received intravitreal injections of either AAV2-d/n REST or AAV2-GFP viruses. After allowing one week for FG transport and viral gene expression in RGCs[46,47,62], retinas were dissected, dissociated with papain, and the dissociated retinal cells were plated on poly-L-lysine pre-coated culture plates. To obtain a baseline of plated RGCs from different retinas, we carried out an initial quantitation of FG-labeled RGC numbers in culture 5–12 h after plating cells. Axon outgrowth and RGC survival were evaluated after 3 days in culture, and each experimental condition was tested in quadruplicate. Counting was carried out using a fluorescent inverted microscope by an observer who was blind to treatment. RGCs were identified by FG labeling under fluorescent illumination, then evaluated for axon growth using phase-contrast to obtain the percentage of RGCs that extended axons

≥30 μm in length. Cell survival is reported as the number of FG-positive RGCs per 40× microscope field averaged over ≥ 30 pre-specified fields per well. The RGC numbers counted at D3 were first normalized by their own initial number at 5–12 h after plating, then averaged within the group. In some cases, cultured cells were immunostained with a rabbit monoclonal antibody to GAP-43 (1:500, Abcam, ab75810) to visualize regenerating axons.

### Dissociated dorsal root ganglion neuronal cultures and neurite outgrowth assay

Adult C57BL/6 J dissociated DRG cells were plated at a concentration of 5,000–10,000 cells/ml in tissue culture plates coated with poly-L-lysine (Invitrogen, 0.1 mg/ml) and laminin (Invitrogen, 2 μg/ml) only or with CSPG (Millipore, 5 μg/ml) cultured in Neurobasal A medium (Invitrogen) containing B27 supplement, penicillin, streptomycin, 1 mM L-glutamine, 50 ng/ml NGF, and 10 mM AraC at 37 °C. REST overexpression was performed by transducing DRG neurons with lentiviral constructs containing either REST (Lv135-REST) or humanized luciferase protein (Lv135-hLuc) as a control driven by the CMV promoter (GeneCopoeia). DRG neurons were replated 7 days after the viral infection. Replated neurons were allowed to grow for another 17–24 h before quantifying neurite outgrowth. To test neurite growth on laminin or CSPG, DRG neurons dissected from REST<sup>flx/flx</sup> mice were dissociated and REST was depleted by infecting neurons with AAV-CRE (experimental) or AAV-GFP (control), the same AAVs used in the Methods section *"Spinal cord injury and corticospinal tract (CST) injections"*, at a viral titer of -100,000 genome copies per cell. Neurite growth was measured after 7 days, and each experimental condition was tested in triplicate. To stain DRG neurites, cells were fixed with 4% paraformaldehyde and blocked for one hour at room temperature in PBS with 0.05% Tween-20 + 0.01% Triton-X + 1% BSA + 5% goat serum, followed by primary antibody incubation with ß-III-tubulin (Biolegend 801201, 1:500) overnight at 4 °C in blocking solutions and secondary antibody (Invitrogen, 1:500) for 1–2 h at room temperature. For quantification of DRG neurites, at least 9 images were randomly taken from each replicate using a Zeiss Confocal Microscope at 20x. Neurites were counted using Imaris Surface Rendering function, and the average neurite surface per neuron was quantified.

### qRT-PCR

RNA from various treatment groups was extracted using the RNeasy kit (Qiagen), reverse-transcribed to cDNA with iScript cDNA Synthesis kit (Bio-Rad) or Quantitect Reverse Transcription kit (Qiagen) for low-input samples. Real-time qPCR was carried out with iTaq Universal (Bio-Rad) or Quantitect (Qiagen) SYBR Green supermix. The primers used in qPCRs were: SPRR1a F: GTCCATAGCCAAGCCTGAAGA; R: GGCAATGGGACTCATAAGCAG; GAP-43 F: GTTTCCTCTCCTGTCCT GCT; R: CCACACGCACCAGATCAAAA. BDNF F: CACTGTCACCTGCTC TCTAGGGA; R: TTTACAATAGGCTTCTGATGTGG; ATF3 F: CTGGG ATTGGTAACCTGGAGTTA; R: TGACAGGCTAGGAATACTGG; REST F: CGACCAGGTAATCGCAGCAG; R: CATGGCCTTAACCAACGACA; 18 S F: CGGCTACCACATCCAAGGAA; 18 S R: GCTGGAATTACCGCGGCT. qPCR data was collected by Roche LightCycler. Relative expression levels in experimental groups were first normalized to those of the reference gene 18 S rRNA, then normalized by the relevant control group depending on the experimental design. Statistical significance among groups was evaluated by one-way ANOVA followed by Bonferroni or Tukey corrections.

### Western blots

Lysates from DRG neurons were run on 4-12% Bis-Tris gradient gels and proteins were transferred to PVDF membranes that were incubated with antibodies to REST (Abcam ab21635, 1:1000), using anti-β-actin as a loading control. Quantitation of western blot results was carried out with ImageJ software.

## FACS isolation of adult cortical motor neurons

Surgeries and AAV injections were carried out in the same way as described in the Methods section "*Spinal cord injury and corticospinal tract (CST) injections*". In order to induce neuron-specific REST depletion, we used AAVs expressing GFP or Cre recombinase under the human synapsin promoter. Adult mouse brain tissue was dissociated as previously described. Briefly, sensorimotor cortex injected with AAV-Syn-GFP or AAV-Syn-CRE to induce tdTomato expression from REST$^{flx/flx}$; tdTomato mice was immediately dissected into ice-cold Hibernate A without calcium (BrainBits, HA − Ca). Tissue was digested by activated papain (Worthington, resuspended in 5 ml HA-Ca) with 100 µl DNase I (2 mg/ml, Roche) in a 37 °C incubator shaking orbitally for 30 min. Digested tissue was triturated gently until clumps disappeared, spun down, and resuspended in 3 ml HA −Ca containing 10% v/v ovomucoid (Worthington, resuspended in 32 ml HA −Ca). Cell debris was removed using discontinuous density gradient containing 3 ml tissue mixture on top of 5 ml ovomucoid solution. Cells were spun down at $70 \times g$ for 6 min and the pellet was resuspended in 1.8 ml Hibernate A low fluorescence (HA-LF; BrainBits) to create a mononuclear cell suspension. Miltenyi myelin removal kit was used to further reduce the amount of debris according to the manufacturer's protocol. Briefly, 200 µl myelin removal beads (Miltenyi) were added to the cell suspension and incubated at 4 °C for 15 min, then the cell suspension was centrifuged at $300 \times g$ for 10 min at 4 °C. The pellet was resuspended in 1 ml of HA-LF and applied to LS columns (Miltenyi) attached to MACS magnetic separator in order to remove beads with myelin. Flow-through, as well as two − 1 ml washes with HA-LF, were collected, centrifuged at $600 \times g$ for 5 min at 4 °C and resuspended in 750 µl HA-LF. Myelin-depleted samples were labeled with live cell marker DRAQ5 (1 µl per sample; Thermo Fisher Scientific) and dead cell marker NucBlue (1 drop per sample; Invitrogen). Samples were FACS-sorted on a Becton Dickinson FACS Aria cell sorter gating for DAPI − /DRAQ5 + /GFP + cells (Supplementary Fig. 10) directly collected in 100 µl of RA1 lysis buffer with 2 µl tris(2-carboxyethyl)phosphine (TCEP) from NucleoSpin RNA XS kit (Clontech).

## FACS isolation of retinal ganglion cells

To investigate the transcriptome of RGCs undergoing axon regeneration, B6.Cg-Tg(Thy1-CFP)23Jrs/J mice, which express cyan-fluorescent protein selectively in RGCs[44], received intraocular injections of either a well characterized adeno-associated virus expressing shRNA against PTEN mRNA[45] and mCherry (AAV2-H1-shPten.mCherry-WPRE-bGHpA, in short: AAV2-shPten.mCherry), or a control virus expressing shLuciferase.mCherry (AAV2-H1-shLuc.mCherry-WPRE-bGHpA, in short: AAV2-shLuc.mCherry). After allowing two weeks for expression of virally encoded genes, mice underwent optic nerve crush. Experimental mice received an intraocular injection of recombinant oncomodulin (rOcm, 90 ng) plus CPT-cAMP (cAMP, 50 µM, total volume = 3 µl); control mice received intraocular saline. At one, three or five days post-surgery, mice were euthanized, retinas were dissected and dissociated by gentle trituration in the presence of papain, and cells were separated by fluorescent-activated cell sorting (FACS, BD Biosciences) on the basis of being positive for both CFP and mCherry (i.e., virally transfected RGCs). We typically obtained 2,000–11,000 RGCs per retina and pooled RGCs from 2 to 3 similarly treated retinas for one sample depending on the number of sorted cells; each condition was repeated at least 8 times in independent experiments.

To investigate the effects of REST manipulations on regeneration-associated TFs and other genes, we injected WT 129S1 mice intravitreally with AAV2-d/nREST (vs. AAV2-GFP in controls) and, at the same time, injected Fluorogold (Fluorochrome) into the superior colliculus (SC) to retrogradely label RGCs. The optic nerve was crushed two weeks later and, after allowing a one week survival period, we euthanized mice, dissected the retinas, dissociated cells (for details see retinal dissociated cell culture) and selected FG-positive RGCs by FACS. RNA from sorted RGCs was extracted for each sample and prepared for real-time qPCR analysis.

## Transcriptional regulatory network analysis

A stepwise pipeline was used to construct a hierarchical TF network from gene expression datasets. Step 1: The Algorithm for Reconstruction of Accurate Cellular Networks (ARACNe)[27] was applied to each of the gene expression profiling datasets to infer directionality among TFs using RTN package[78]. Pair-wise mutual information (MI) scores were computed and non-significant associations were removed by permutation analysis (permutation = 100; FDR adjusted $p$ value < 0.05; consensus score = 95%). Unstable interactions were removed by bootstrapping, and indirect interactions such as two genes connected by intermediate steps were removed by data-processing inequality (DPI) of the ARACNe algorithm. Step 2: To further confirm the directionality inferred by ARACNe, we examined evidence of physical TF-target binding observed by multiple ChIP-Seq or ChIP-ChIP databases[13,33]. Step 3: To define the hierarchical structure of the directed TF network, we used a graph-theoretical algorithm called *vertex-sort* (16, code in Supplementary Information), which essentially assign TFs into a multi-layer structure based on their connectivity statistics. Compared to other similar algorithms to infer network hierarchy[79–82], *vertex-sort* considers cyclic sub-network structure (e.g. directed path from u to v, v to u) by collapsing them into super-nodes. It applies the iterative leaf-removal algorithm to a directed network and to its transpose, which begins by finding the bottom-level TF nodes in a directed network, including both the TFs that do not regulate others and those that are only self-regulating; TFs that directly regulate the bottom-level genes are then pushed the next level, and this process is repeated until all TFs are assigned to a specific level. Finally, the normal and the transposed networks are combined obtain the final topological order of TF nodes. This allows for an approximate stratification of TFs within each dataset, without probability of a node at a specific level, which is a limitation of the *vertex-sort* algorithm. Edges and nodes in the network were visualized by igraph R package. Centrality statistics of each TF node was calculated using qgraph R package *centrality_auto ()* function.

## Network motif analysis

mFinder v1.21 software was used for network motif analysis[83]. A Z-score was calculated for each of 13 network motifs with 3-node structure, using 1000 random networks of the same size for background estimation.

## RNA-seq library preparation

RNA from FACS-sorted neurons of the sensorimotor cortex (~1000 cells) was isolated with the NucleoSpin RNA XS kit (CloneTech) according to the manufacturer's protocol. RNA-seq libraries for cortical motor neurons were prepared with the QuantSeq 3'mRNA-Seq library prep kit FWD for Illumina (Lexogen) following the manufacturer's instructions, while RNA-seq libraries for RGCs were generated using TruSeq with RiboZero gold following the manufacturers's instructions. The cDNA was fragmented to ≈300 base pairs (bp) using the Covaris M220 (Covaris), and then the manufacturer's instructions were followed for end repair, adaptor ligation, and library amplification. The libraries were quantified by the Qubit dsDNA HS Assay Kit (Molecular Probes); Library size distribution and molar concentration of cDNA molecules in each library were determined by the Agilent High Sensitivity DNA Assay on an Agilent 2200 TapeStation system. Libraries were multiplexed into a single pool and sequenced using a HiSeq4000 instrument (Illumina, San Diego, CA) to generate 69 bp single-end reads. The average read depth for each library is ~11 million for cortical motor neurons and ~33 million for RGCs.

### RNA-seq read alignment and processing

Sensorimotor cortex neuronal RNA-seq data were mapped to the reference genome (mm10/GRCm38) using STAR[84]. Alignment and duplication metrics were collected using PICARD tools functions CollectRnaSeqMetrics and MarkDuplicates respectively (http://broadinstitute.github.io/picard/). Transcript abundance from aligned reads were quantified by Salmon[85], followed by summarization to the gene level using the R package Tximport[86]. Sequencing depth was normalized between samples using geometric mean (GEO) in DESeq2 package[87]. Removal of unwanted variation (RUV) was used to remove batch effects[88] and genes with no counts in over 50% of the samples were removed.

### Gene set enrichment analysis (GSEA)

GSEA v2.0 software with default settings[48] was used to identify upstream TFs of the genes associated with the combined pro-regenerative treatments of AAV2-sh.*pten*, Oncomodulin plus CPT-cAMP. These genes were ranked by their correlations of expression changes with treatments measured by directional p-value, which is calculated as -sign(log Treatment/Control)*(log10 p-value). A positive correlation indicates up-regulation of a gene by pro-regenerative treatment, while a negative correlation indicates down-regulation. A total of 1137 gene sets known to be targeted by transcription factors were downloaded from MsigDB (v5.1), and each set of the TF target genes were compared to the genes associated with the pro-regenerative treatments. An enrichment score (ES) is returned for each comparison, which represents the degree to which the TF-target list is over-represented at the top or bottom of the ranked gene list. The score is calculated by walking down the gene list, increasing a running-sum statistic when we encounter a gene in the TF-target list and decreasing when it is not. The magnitude of the increment depends on the gene statistics so as to determine whether a specific set of a TF's target genes is randomly distributed throughout genes of interest, or primarily found at the top or bottom.

### Differential gene expression

Principle component analysis (PCA) of the normalized expression data (first five PCs) was correlated with potential technical covariates, including sex, aligning and sequencing bias calculated from STAR and Picard respectively. Differential gene expression by limma voom[89] was performed on normalized gene counts, including the first two PCs of aligning and sequencing bias as covariates: ~ Genotype + Align-Seq.PC1 + AlignSeq.PC2. Differentially expressed genes were determined at FDR p value < 0.1 (Supplementary Data 1). Gene overlap analysis between DEGs and REST targeted gene sets was performed using the R package GeneOverlap. One-tailed P values were used (equivalent to hypergeometric P value) since we do not assume enrichment a priori.

### Gene Ontology Analysis

GO term enrichment analysis was performed using the gProfileR package[90] and Ingenuity Pathway Analysis (IPA) Software (Qiagen), using expressed genes in each of the normalized dataset as background. A maximum of top 10 canonical biological pathways, disease and function from each analysis were chosen from GO terms with FDR of p values < 0.05 and at least 10 genes overlapping the test data. The R package clusterProfiler[91] was used to plot the DEGs connecting to a specific GO term, with source code modified to accept GO terms from gProfileR and IPA.

### Weighted gene co-expression network analysis

Sequencing and aligning covariates were regressed out from normalized expression data using a linear model. Co-expression network was constructed using the WGCNA package[43]. Briefly, pair-wise Pearson correlations between each gene pair were calculated and transformed to a signed adjacency matrix using a power of 10, as it was the smallest threshold that resulted in a scale-free $R^2$ fit of 0.8. The adjacency matrix was used to construct a topological overlap dissimilarity matrix, from which hierarchical clustering of genes as modules were determined by a dynamic tree-cutting algorithm (Supplementary Data 3).

### WGCNA module annotation

To classify up- or down-regulated modules, the module eigengene, defined as the first principle component of a module that explains the maximum possible variability of that module, was related to genotype (AAV-Syn-Cre vs AAV-Syn-GFP) using a linear model. Modules were considered to be significantly associated with the phenotype when Bonferroni corrected p values are less than 0.05. A positive association indicates up-regulation of this module comparing neurons expressing AAV-Syn-CRE to the ones expressing AAV-Syn-GFP, while a negative correlation indicates down-regulation. As a first step towards functional annotation, a hypergeometric analysis was used to examine each module's association with the regeneration-associated gene (RAGs) module known to be activated by peripheral injury[8]. Modules were considered to be significantly associated with the RAG program when Bonferroni corrected *p* values are less than 0.05. To further annotate modules at a general level, we applied gene ontology (GO) enrichment analyses on each module. We also calculated Pearson correlations between each gene and each module eigengene as a gene's module membership (Supplementary Data 3), and hub genes were defined as being those with highest correlations (kME > 0.7), which represent the most central genes in the co-expression network.

### Protein-protein interaction (PPI) network analysis

We established interactions of proteins encoded by genes from each of the co-expression modules (RESTUP1 [202 genes], RESTUP3 [636 genes], and RAG module [286 genes]) using InWeb database (https://inbio-discover.com), which combines reported protein interactions from MINT, BIND, IntAct, KEGG annotated protein-protein interactions (PPrel), KEGG Enzymes involved in neighboring steps (ECrel), and Reactome[92,93]. The significance of PPIs within the network was further determined by DAPPLE (https://www.genepattern.org/), which uses a within-degree within-node permutation method that allows us to rank PPI hubs by P value. The PPI networks were visualized by igraph R package (https://igraph.org/r/), or Ingenuity Pathway Analysis (IPA) Software (Qiagen).

### Footprinting analysis

ATAC-seq data generated from uninjured and optic nerve injured RGCs at day 1 and 3 (GSE184547) were used for footprint analysis as described[51]. This ATAC-seq data, with an average of ~90 M unique non-mitochondria read pairs per sample, identified a reproducible set of 151,630 peaks across 15 samples (*N* = 3–6 replicates per condition). For footprinting analysis, we merged biological replicates from each condition by Picard Tools and downsampled ATAC-seq BAMs to a depth of 90 million reads using samtools. The first step of footprinting is to correct Tn5 transposase cleavage bias in the ATAC-seq data. TOBIAS[52] ATACorrect module (default parameters used) was applied to merged reads from biological replicates within consensus peak regions to estimate the background bias of Tn5 transposase. Subtracting the background Tn5 insertion cuts from the uncorrected signals yields corrected signals, highlighting the effect of protein binding. The TOBIAS ScoreBigWig module[52] was used to identify and score DNA footprints corrected ATAC-seq signals within peaks. The footprint score measures both accessibility and depth of the local footprint, thus correlating with the presence of a TF at its target locus, and the chromatin accessibility of the regions where this TF binds. To match footprints to potential REST binding sites, we integrated REST motif PWMs from multiple sources (JASPAR2016, HOCOMOCCO v10, UniPROBE and SwissRegulon), and performed motif enrichment analysis to

identify REST binding sites in footprinted, accessible chromatin regions. Genome browser views of REST footprinted regions were plotted by Gviz[94] on the footprint scores generated by the TOBIAS ScoreBigWig module.

## Statistical analysis

Assumptions concerning the data normality and similar variation between experimental groups were assessed for appropriateness before statistical tests were conducted. Statistical analysis was carried out using one-way or two-way ANOVA with multiple comparisons *post hoc* test, unpaired two-tailed Student's t-test as indicated in the figure legends. Mice with different litters, body weights and sexes were randomized and assigned to different treatment groups, and no other specific randomization was used for the animal studies.

## Reporting summary

Further information on research design is available in the Nature Research Reporting Summary linked to this article.

## Data availability

RNA-seq data generated in this paper have been deposited in the Gene Expression Omnibus (GEO) with accession numbers GSE141583 and GSE142881. ATAC-seq data for footprinting analysis is part of our other study with accession number GSE184547. Source data are provided with this paper.

## Code availability

There was no custom code development and all software used in the current study are published, open-access, and cited under the relevant method sections. Readers can access the code in the software repositories and documentations via the links provided below: RTN (https://bioconductor.org/packages/release/bioc/vignettes/RTN/inst/doc/RTN.html); mFinder (https://www.encodeproject.org/software/mfinder/); RNA-seq (https://github.com/icnn/RNAseq-PIPELINE); tximport (https://bioconductor.org/packages/release/bioc/html/tximport.html); DESeq2 (https://bioconductor.org/packages/release/bioc/html/DESeq2.html); RUV (https://bioconductor.org/packages/release/bioc/html/RUVSeq.html); limma (https://bioconductor.org/packages/release/bioc/html/limma.html); WGCNA (https://horvath.genetics.ucla.edu/html/CoexpressionNetwork/Rpackages/WGCNA/); gProfileR (https://cran.r-project.org/web/packages/gprofiler2/index.html); igraph (https://igraph.org/r/); TOBIAS (https://github.com/loosolab/TOBIAS).

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

## Acknowledgements

We are grateful for the support of the Dr. Miriam and Sheldon G. Adelson Medical Research Foundation (D.H.G., L.B., C.J.W., Z.H., M.S.), the National Eye Institute (U01EY027261-01 to L.B., J.L.G.), U.S. Department of Defense (CDMRP W81XWH-16-1-0043 to L.B., J.L.G.), Wings for Life Spinal Cord Research Foundation (Y.C.). We thank the IDDRC Cellular Imaging Core, funded by NIH P50 HD105351 and S10OD016453 to Boston Children's Hospital. We thank the UCLA BSCRC flow cytometry core for its cell sorting service. We also wish to thank Dr. Gail Mandel (Oregon Health Sciences Univ.), Drs. Mihaela Stavarache and Michael Kaplitt (Weill Cornell Medical College) for generously providing viral vectors and advice, and Dr. Jenny Hsieh (University of Texas at San Antonio) for providing the initial RESTflx/flx breeder mice.

## Author contributions

Y.C., Y.Y., M.V.S., L.I.B and D.H.G. designed and directed the experiments and guided the analysis. Y.C., Y.Y., L.I.B and D.H.G. prepared the figures and wrote the manuscript. Y.C. performed bioinformatic analyses on the RNA-seq datasets of cortical motor neurons and RGCs. A.Z. performed transcription factor network analysis on the microarray datasets. Y.C. and A.Z. performed experiments on mouse DRG cultures with guidance from C.J.W. Y.Y. performed experiments on mouse RGC cultures. Y.C., A.M.B., K.G., Y.A., and K.P. performed SCI experiments and collected brain and spinal cord samples for immunostaining. Y.C., K.G. and J.O. processed cortical motor neurons and prepared RNA-seq libraries. Y.Y. and H.Y.G. conducted optic nerve crush experiments, processed retina for immunostaining and RGCs for RT-PCR and RNA-seq. R.K. performed initial processing of RNA-seq data from RGCs. Y.C., A.Z., and C.J.K. bred the REST^flx/flx and REST^flx/flx; STOP^flx/flxTdTomato mice. Y.C. performed REST foot-printing analysis of RGC ATAC-seq with advice from Z.H. All authors discussed the results and provided comments and revisions on the manuscript.

## Competing interests

Z.H. is a co-founder of Rugen and Myro Therapeutics. The remaining authors declare no competing interests.
