## [Peer Review File · Nature Communications]

Reviewers' Comments:

Reviewer #1:

Remarks to the Author:

The authors have discussed the points raised adequately and have provided some additional data and analysis.

The impact of REST deletion upon the growth of corticospinal axons following SCI remains very underwhelming, affecting retraction mainly. The choice of a different SCI model leaving some astrocytic bridges would have allowed a more appropriate assessment of axonal sprouting, regeneration as well as retraction.

Reviewer #2:

Remarks to the Author:

In this manuscript the authors execute a number of functional experiments that provide solid and novel evidence implicating REST as a negative upstream regulator of CNS axon regeneration. The experiments are logically laid out and the results are fairly clear.

I nonetheless have some major reservations about the manuscript related to their analytical approaches and their use of chromatin data. They claim to have performed an "unbiased analysis" implicating REST as a prelude to their experimental studies, and repeat the word "unbiased" throughout the manuscript to characterize their approach, which is in fact heavily biased to a particular class of result. While the results of their network analysis may have been the chief motivation for their experimental analyses focusing on REST, the success of the latter does not justify the former; even a broken clock that tells the right time twice a day. Further, the authors have introduced a new Figure 5 purporting show direct REST occupancy on the basis of chromatin accessibility data. This analysis is deeply flawed.

I will focus my comments on these two aspects of the manuscript, both of which either need major revision or could be removed without impact on the central findings, which derive from the results presented in Figures 3-4 and 6-8.

1) Network analyses

Despite the central role claimed for the network analysis, it is very scantily described in only one relatively short paragraph in the Methods that chiefly refers to other papers. In outline:

- The authors use the well-established ARACNE network inference approach, applied to third-party microarray data from diverse sources. Notably, data from the peripheral nervous system are twice as abundant as those from the CNS, which will have a major impact on network accuracy and robustness.
- They then apply a little-used heuristic (Jothi et al., 2009) that effectively purges the regulatory network of complex network motifs by stripping out loops. Essentially, what they have done is to take a network and then REMOVE everything that is not a hierarchical relationship. In effect, they have converted a rich network into one in which mostly feed-forward loops remain. Notably, because REST is known to control a large regulon, it will be positioned as the superior node in many feed forward loops.
- By stripping out other network architectures, it is highly probable that the authors have indirectly enriched for REST-containing loops. They do not attempt to test or correct for this possibility, nor even is it made clear in the text or methods that this is what they are doing.

- Following purging of loops, the network has largely been reduced to three layers, which they authors claim is a natural hierarchy. They point to a series of papers from the same lab (Gerstein, Yale) who have performed a similar approach to theirs.
- The hierarchical arrangement claimed is not supported by other analyses of the TF network, and flies in the face of the fundamental results on network motifs from Alon and others.
- The authors make an additional mistake by using data from ENCODE to filter their results. It is a well known limitation that the ENCODE transcription factor chip-seq data derive almost from two human cell lines (K562 erythroleukemia and HepG2 hepatoblastoma), neither of which has anything to do with neuronal cells or tissues - a critical deficit since the differences in network organization and behavior are expected to be large between different cell types.
- The purported differences the authors claim to show between the PNS and CNS network architectures shown in Figure 2 are totally non-quantitative.

Thus, far from an unbiased approach, the imbalance input data sources, the purging of network loops, the filtering of results for non-neuronal cell types, the forcing of the network into feed-forward 3-layer structures imposed by the authors' methods (which are non-transparent), and the lack of quantitative network comparisons make for a very muddled picture.

2) Direct interactions of REST (new Figure 5).

The authors have added a new section that purports to demonstrate direct REST occupancy on the genome. I find this analysis, its description, the relevant figure panels, and the supporting methods to be unintelligible.

- Figure 5 purports to show REST occupancy but actually shows no such thing, and many the panels are missing key labels.
- Figure 5B is described as showing "Aggregated footprint signals" +/-2kb from the TSS of genes purportedly bound by REST. But the figure actually shows some kind of peak representation in the center of a 200bp interval.
- Figure 5C is missing any indication of the number of genes studied. It also shows an entirely expected relationship between promoter accessibility and gene expression that is fundamental and has nothing to do with REST per se.
- Figure 5D purports to show "mean accessibility of REST's binding sites". What it contains are a number of peaks with redacted data in between, and no indication of where the REST binding sites are supposed to be or how they were determined.

The methods accompanying the new text section and figure are contained in a short paragraph devoid of any details of how REST binding sites were determined, nor are any details of the experiment reported including even basic parameters such as numbers of mapped reads. Furthermore, the authors cite a data set associated with Figure 5 and state that it is "manuscript to be submitted". So are the data part of the present paper or not?

Summary

In summary, this is a paper with solid functional experimental data showing, empirically, that REST negatively regulates peripheral axon regeneration. The paper is prefaced by a forced and problematic computational analysis that does not contribute any real insight; the entire purpose of this analysis is to provide a justification for investigating REST. Whether REST was fingered by this analysis is immaterial to the actual experiments performed. The paper also contains an analysis purporting to show that REST directly interacts with target genes by using footprinting; this analysis is impenetrable and unconvincing.

The authors could simplify their presentation greatly by stating accurately and transparently what they did analytically to develop circumstantial evidence implicating REST, without claiming either a lack of bias or some generalizable approach, and truncating the early figures correspondingly. Figure 5 could be dropped entirely.

Reviewer #3:

Remarks to the Author:

The authors have addressed my concerns, I have no further issues.

RESPONSE TO REVIEWER COMMENTS

Reviewer #1 (Remarks to the Author):

The authors have discussed the points raised adequately and have provided some additional data and analysis

Thank you for your guidance.

The impact of REST deletion upon the growth of corticospinal axons following SCI remains very underwhelming, affecting retraction mainly. The choice of a different SCI model leaving some astrocytic bridges would have allowed a more appropriate assessment of axonal sprouting, regeneration as well as retraction.

We agree with the Reviewer that the more axons present in REST-deficient mice could be either a lack of dieback in the axons or a regrowth of axons after anatomically complete spinal cord injury. To distinguish between these potential mechanisms, we first examined CST axons 3 days post-injury. Apparent dieback and large numbers of retraction bulbs were observed at this early time point in both control and REST-deleted axons (Original Fig. S8A), indicating that REST depletion promotes regenerative axon growth, rather than preventing dieback.

We also measured branching of CST axons at 4 weeks post injury which, when increased, is considered to be strong evidence of regenerative growth (e.g. Sofroniew et al., 2018) (Original Fig. 7D). Mice receiving AAV-Cre displayed far more branching from injured CST axons in the area proximal to the lesion center than controls (Fig. 7E-F), further supporting that REST deletion enhances axon regeneration.

We absolutely agree that using an incomplete SCI model to assess axonal sprouting and regeneration would be of clinical relevance and should be investigated in the future. However, we want to clarify that a major goal here is to validate a *model* that our bio-informatic findings predict, that is, that REST acts as a transcriptional repressor of CNS neuronal growth. This prediction is supported by several in depth experiments including the complete SCI model and the ON crush model, as well as the transcriptional analysis of REST-depleted, CNS-injured neurons.

Reviewer #2 (Remarks to the Author):

In this manuscript the authors execute a number of functional experiments that provide solid and novel evidence implicating REST as a negative upstream regulator of CNS axon regeneration. The experiments are logically laid out and the results are fairly clear.

We thank the reviewer for acknowledging the clarity of the functional experiments and Results.

I nonetheless have some major reservations about the manuscript related to their analytical approaches and their use of chromatin data. They claim to have performed an “unbiased analysis” implicating REST as a prelude to their experimental studies, and repeat the word “unbiased” throughout the manuscript to characterize their approach, which is in fact heavily biased to a particular class of result. While the results of their network analysis may have been the chief motivation for their experimental analyses focusing on REST, the success of the latter does not justify the former; even a broken clock that tells the right time twice a day. Further, the authors have introduced a new Figure 5 purporting show direct REST occupancy on the basis of chromatin accessibility data. This analysis is deeply flawed.

We address this reviewer’s reservations below and in the revised manuscript. In the revision, we have removed the term “unbiased” throughout. Further, to clarify our approach, we provide a more detailed explanation of the TF network analyses, as well as clarifications on the REST footprinting analysis, as described below. We respectfully disagree that our network approach is heavily based towards a particular class of result, specifically, the feed-forward loop motifs as mentioned by the Reviewer. Globally, the 3-layer TF network does resemble a feed-forward structure, but locally, each layer actually does contain different types of network motifs, in agreement with previous studies by Alon and others. Therefore, there is no bias towards a certain type of motif structure in this analysis. We also recognize that the ENCODE database does not contain neuronal ChIP-seq data, but that does not eliminate its value – no database is complete and we go on to validate specific predictions in the neurons. Because certain cell types are missing from the databases, and no analysis is completely unbiased, we have taken this reviewer’s guidance and removed ‘unbiased’ in the manuscript as mentioned above.

We agree with the Reviewer that our experimental work does not necessarily prove a 3-layer structure of the TF network, which is based on a published model (see also below). However, it fits with this model and in the later part of the manuscript, both RNA-seq (Fig.3B) and immunohistochemistry (Fig.8G-I) data from CNS-injured neurons demonstrate that the inhibition of REST increases the core regenerative TFs (ATF3, STAT3, SMAD1, JUN), which is a prediction of the network analyses. As the current and previous Reviewers point out, and as was actually the sequence of events that drove these experiments in our labs, these network analyses laid the foundation for the hypothesis testing that follows. Without the bioinformatic predictions we would not have performed our in-depth experiments involving optic nerve and spinal cord injury, nor would we identify a new role for REST in axon regeneration. Thus, the network analyses are an indispensable part of this manuscript.

We apologize for not providing sufficient details on the DNA footprinting analysis to predict REST occupancy on the chromatin. We included clarifications and explanations in the manuscript, as described below.

I will focus my comments on these two aspects of the manuscript, both of which either need major revision or could be removed without impact on the central findings, which derive from the results presented in Figures 3-4 and 6-8.

1) Network analyses

Despite the central role claimed for the network analysis, it is very scantily described in only one relatively short paragraph in the Methods that chiefly refers to other papers. In outline:

- The authors use the well-established ARACNE network inference approach, applied to third-party microarray data from diverse sources. Notably, data from the peripheral nervous system are twice as abundant (sic) as those from the CNS, which will have a major impact on network accuracy and robustness.

We agree with the Reviewer – less data could lead to missing observations. However, we have used all possible good quality, high-density time course microarray data that we could find in the CNS. Additionally, we do not see REST or CTCF “interrupting” the network in the PNS, which should be more powered to detect a more complete set of interactions. To add more confidence, we have generated another RNA-seq data set to compare CNS neurons with or without pro-regenerative interventions (4 time points, N= 5-7 replicates per condition). REST was also found in the top-level of the network in the non-regenerative CNS neurons (Fig. 4E), similar to the CNS microarray in Fig. 2. These consistent findings across multiple data sets encouraged us to validate the prediction of REST being a transcriptional repressor in animal models of CNS regeneration. We also go on to provide experimental evidence that REST binds to the promoters of regeneration associated genes, and indeed down-regulates them, in addition to the experiments showing improved regeneration with REST reduction.

- They then apply a little-used heuristic (Jothi et al., 2009) that effectively purges the regulatory network of complex network motifs by stripping out loops. Essentially, what they have done is to take a network and then REMOVE everything that is not a hierarchical relationship. In effect, they have converted a rich network into one in which mostly feed-forward loops remain.

We thank the Reviewer for these comments. We would like to clarify the hierarchical network analysis we performed (Jothi et al., 2009) to address the Reviewer’s concerns.

First, the vertex-sort algorithm described by Jothi et al., 2019, as well as other similar methods to characterize the hierarchical structure of a transcriptional regulatory network (TRN) (Ma et al, 2004a, 2004b; Balazsi et al, 2005; Yu and Gerstein, 2006), are widely supported by studies in multiple biological processes of multiple organisms (Bhardwaj et al., 2010a; Bhardwaj et al., 2010b; Gerstein et al., 2012; Song et al., 2016). These algorithms essentially assign TFs into a multi-layer structure based on their connectivity statistics. They begin by finding the bottom-level nodes in a directed network, *including both* the TFs that do not regulate others and those that are only self-regulating; TFs that directly regulate the bottom-level genes are pushed the next level, and this process is repeated until all TFs are assigned to a specific level.

Compared to other similar algorithms, vertex-sort is scalable, and considers cyclic sub-network structure (e.g. directed path from u to v, v to u). In contrast to the reviewer’s statements above, by searching the topological orderings in both bottom-up and top-down manners, **this method tries to keep all possible regulations including the non-hierarchical ones.** This method neither forces the network into a feed-forward structures. Jothi et al., describes a total of 7 layers in the yeast TRN, which are clustered into 3 non-overlapping layers – the top, the core/middle, and the bottom layer, resembling a feed-forward structure. This global structure is different from the individual feed-forward loop motifs studied by Uri Alon and others (Milo et al., 2002, Alon 2007), which is further discussed under our response to the Reviewer’s next comment.

Prior to building the hierarchical structure, the ‘removing’ step occur during ARACNE analysis, specifically on those TF-gene interactions that are less stable based on a bootstrapping analysis. The parameters (permutation = 100; FDR adjusted p value < 0.05; consensus score = 95%) used for the permutation analysis were described in the original manuscript. Following ARACNE, the TF-

gene interactions were further filtered by the *experimental* ChIP-seq data from ENCODE and other ChIP-ChIP databases (Landt et al., 2012, Lachmann et al., 2010). However, neither of these analyses would specifically strip out non-hierarchical regulatory relationships.

- Notably, because REST is known to control a large regulon, it will be positioned as the superior node in many feed forward loops.

- By stripping out other network architectures, it is highly probably that the authors have indirectly enriched for REST-containing loops. They do not attempt to test or correct for this possibility, nor even is it made clear in the text or methods that this is what they are doing.

As described above, we did not strip out non-hierarchical network architecture. It is also unlikely we enriched REST-containing loops. As described in the Methods, the network construction by ARACNE is entirely based on the gene-expression patterns themselves, mitigating this concern. In addition, we applied the same framework to the same set of TFs in both PNS and CNS injury datasets, yet REST and CTCF were only found to interact with other TFs in the CNS but not the PNS, indicating that our analysis does not depend on whether the TFs have large regulons or not.

- Following purging of loops, the network has largely been reduced to three layers, which they authors claim is a natural hierarchy. They point to a series of papers from the same lab (Gerstein, Yale) who have performed a similar approach to theirs.

- The hierarchical arrangement claimed is not supported by other analyses of the TF network, and flies in the face of the fundamental results on network motifs from Alon and others.

The 3-layer hierarchy of TRN does not conflict with the network motif concept. In fact, this global structure contains all possible formats of these small recurring regulatory patterns, among which the most enriched type is the feed-forward loop motifs (Gerstein et al., 2012; Jothi et al., 2009). This actually is in agreement with studies from Alon's group, which found feed-forward loop and bi-fan motifs to be the most abundant patterns in yeast and bacteria in earlier studies (e.g. Milo et al., 2002).

- The authors make an additional mistake by using data from ENCODE to filter their results. It is a well known limitation that the ENCODE transcription factor chip-seq data derive almost from two human cell lines (K562 erythroleukemia and HepG2 hepatoblastoma), neither of which has anything to do with neuronal cells or tissues - a critical deficit since the differences in network organization and behavior are expected to be large between different cell types.

We agree with the Reviewer that ENCODE does not use neuronal cells. We included this limitation in the Discussion But, many (often 50% or more) regulatory relationships are conserved across different cell types and indeed we validate the predictions made in these non-neural cells in multiple subsequent experiments in vivo in neurons.

- The purported differences the authors claim to show between the PNS and CNS network architectures shown in Figure 2 are totally non-quantitative.

We apologize for this omission. We now included the following network statistics to support the difference we found between the PNS and CNS network. As a general view of the network connectivity, we first calculated betweenness centrality, and in- and out-degree for the top, middle, and bottom layer of each PNS or CNS network. As expected, we observed higher betweenness for middle layer TFs, as they connect top and bottom layer TFs (**graph below**). The out-degree for TFs in different layer is top > middle > bottom, as the top-layer TF mainly regulates others while the bottom-layer TFs receives regulation. No obvious in-degree difference is observed among layers.

One major difference we observed in the CNS and PNS network structure is that CNS networks are less interconnected. To quantitate this observation, we calculated global and local clustering coefficient, the former indicating the global network connectivity, while the latter informing local connectivity of each TF node. We observed lower global clustering coefficient in the CNS networks compared to the PNS ones (**new Fig. S1A, below, globalCC**). Likewise, TF's local connectivity is lower in the CNS network in general (**new Fig. S1A, localCC**).

Another observation is that the five TF subnetwork consisting of ATF3, JUN, STAT3, SOX11, SMAD1 appear to be preserved across the PNS injury datasets, but not in the CNS. We had quantitated the betweenness, in- and out-degree for these 5, but they do not seem to be informative due to relatively small size of each TF network (**graph below**). Alternatively, we calculated the similarity of their regulons across PNS and CNS datasets. In the original **Fig. S1C**, we showed that Atf3, Jun, Sox11, and Smad1 bear the most correlated regulatory relationships with each other across multiple PNS injury datasets. By contrast, there is little correlation in the regulatory interactions of the core TFs between PNS and CNS injury datasets, providing additional support.

Betweenness, in- and out-degree for CTCF, REST, SMAD1, JUN, STAT3, SOX11 and ATF3.

-Thus, far from an unbiased approach, the imbalance input data sources, the purging of network loops, the filtering of results for non-neuronal cell types, the forcing of the network into feed-forward 3-layer structures imposed by the authors' methods (which are non-transparent), and the lack of quantitative network comparisons make for a very muddled picture.

As described above, we had included another dataset comparing non-regenerative to pro-regenerative CNS neurons for the same network analysis in the original manuscript (Fig. 4E), demonstrating similar findings in the microarray data; We included more details describing the TF network analysis, which do not force a feed-forward structure. We provided several quantitations of the global network structure and the comparison between PNS and CNS networks. We do agree with the Reviewer that using non-neuronal cell types is a limitation and does not justify the approach being 'unbiased'. Therefore, we removed this phrase from the manuscript where related.

2) Direct interactions of REST (new Figure 5).

The authors have added a new section that purports to demonstrate direct REST occupancy on the genome. I find this analysis, its description, the relevant figure panels, and the supporting methods to be unintelligible.

- Figure 5 purports to show REST occupancy but actually shows no such thing, and many the panels are missing key labels.

To clarify, we used TF footprinting analysis of ATAC-seq data from another submitted manuscript to predict REST occupancy (Tian et al., 2022), which is now cited. ATAC-seq measures chromatin accessibility via transposase (Tn5), which recognize and cleave DNA in open chromatin. TF-bound DNA in open chromatin is protected from enzymatic cleavage, leaving small DNA regions as 'footprints', which can be used to predict core TF binding sites (**Fig.5A, below**). To match DNA footprints to potential REST binding sites, we overlapped DNA footprints with REST motif position-weight-matrixes from multiple sources (JASPAR2016, HOCOMOCCO v10, UniPROBE and SwissRegulon). This analysis has been used by us and others to identify TF binding sites, comparable to, but with higher resolution than ChIP-seq (Li et al., 2019; Bentson et al., 2020, Vierstra et al., 2020; Pierce et al., 2021; Cheng et al., 2021; Tian et al., 2022). Because it is technically not feasible to generate high-quality REST ChIP-seq in purified, adult primary neurons from CNS-injured animals, we took this alternate approach to leverage ATAC-seq generated in another manuscript for another purpose to identify REST binding sites (Tian et al., 2022).

- Figure 5B is described as showing "Aggregated footprint signals" +/-2kb from the TSS of genes purportedly bound by REST. But the figure actually shows some kind of peak representation in the center of a 200bp interval.

We apologize for not being clear. The description meant that the promoter regions of a gene is defined as +/-2kb from the TSS of genes. The Aggregated footprint signals are plotted +/-100 bp (200 bp interval) on either side of the center of REST binding sites. We realized that the footprint plot provides similar information as in the heatmap of Fig. 5B, demonstrating decreased accessibility surrounding REST footprinted promoter regions. To be clearer and more concise, we removed the footprint plot and kept the heatmap plot, which is described in the response to Reviewer's next comments below.

- Figure 5C is missing any indication of the number of genes studied. It also shows an entirely expected relationship between promoter accessibility and gene expression that is fundamental and has nothing to do with REST per se.

We agree with the Reviewer that the correlated promoter accessibility and gene expression is not necessarily driven by REST binding. We removed the correlation graph and kept the heatmap (new Fig. 5B) to demonstrate decreases in both promoter accessibility and mRNA levels for most REST-targeted genes. These results, together with the findings of increased expression of regenerative TFs upon REST inhibition (Figure 3B; Figure 8 G-I), provide strong evidence that REST directly binds to and represses expression of TFs that would otherwise drive axon regeneration. We also described in the text that a total of 801 genes were found downstream of promoter regions that overlap with REST footprints as potential REST binding sites.

- Figure 5D purports to show “mean accessibility of REST’s binding sites”. What it contains are a number of peaks with redacted data in between, and no indication of where the REST binding sites are supposed to be or how they were determined.

We thank the Reviewer for this comment. To clarify, the peaks shown are REST footprinted regions predicted by the TF footprinting analysis described above. The footprint signals are TF binding scores that calculates both accessibility and depth of each footprint (**Fig.5A, above**), correlating with the presence of a TF at its target loci, and the chromatin accessibility of the regions where this TF binds. To be more accurate, we changed ‘mean accessibility of REST binding sites’ to ‘mean footprint scores’ of predicted REST binding sites, which is the average of REST footprint scores across the entire gene. These details are described in the new **Fig.5 legend below** and Methods.

- The methods accompanying the new text section and figure are contained in a short paragraph devoid of any details of how REST binding sites were determined, nor are any details of the experiment reported including even basic parameters such as numbers of mapped reads. Furthermore, the authors cite a data set associated with Figure 5 and state that it is “manuscript to be submitted”. So are the data part of the present paper or not?

We are sorry for this confusion. We mentioned in Data Availability that the “ATAC-seq data for footprinting analysis is part of our other study (manuscript under preparation) with data accession number GSE184547”. This manuscript is now in bioRxiv (Tian et al., bioRxiv 2022.01.20.477004; doi: <https://doi.org/10.1101/2022.01.20.477004>).

We now added more details in the Methods section under ‘Footprinting analysis’, highlighted in red below:

Footprinting analysis. ATAC-seq data generated from uninjured and optic nerve injured RGCs at day 1 and 3 were used for footprint analysis (GSE184547) as described (Tian et al., *bioRxiv* 2022.01.20.477004; doi: <https://doi.org/10.1101/2022.01.20.477004>). This ATAC-seq data, with an average of ~90M unique non-mitochondria read pairs per sample, identified a reproducible set of 151,630 peaks across 15 samples (N = 3-6 replicates per condition). For footprinting analysis, we merged biological replicates from each condition by Picard Tools and downsampled ATAC-seq BAMs to a depth of 60 million reads using samtools. The first step of footprinting is to correct Tn5 transposase cleavage bias in the ATAC-seq data. TOBIAS (51) ATACCorrect module (default parameters used) was applied to merged reads from biological replicates within consensus peak regions to estimate the background bias of Tn5 transposase. Subtracting the background Tn5 insertion cuts from the uncorrected signals yields corrected signals, highlighting the effect of protein binding. The TOBIAS ScoreBigWig module (51) was used to identify and score DNA footprints corrected ATAC-seq signals within peaks. The footprint score measures both accessibility and depth of the local footprint at a 20-50 bp range, thus correlating with the presence of a TF at its target locus, and the chromatin accessibility of the regions where this TF binds. To match footprints to potential REST binding sites, we integrated REST motif PWMs from multiple sources (JASPAR2016, HOCOMOCCO v10, UniPROBE and SwissRegulon), and performed motif enrichment analysis to identify REST binding sites in footprinted, accessible chromatin regions. Genome browser views of REST footprinted regions were plotted by Gviz (95) on the footprint scores generated by the TOBIAS ScoreBigWig module.

Summary

In summary, this is a paper with solid functional experimental data showing, empirically, that REST negatively regulates peripheral axon regeneration. The paper is prefaced by a forced and problematic computational analysis that does not contribute any real insight; the entire purpose of this analysis is to provide a justification for investigating REST. Whether REST was fingered by this analysis is immaterial to the actual experiments performed. The paper also contains an analysis purporting to show that REST directly interacts with target genes by using footprinting; this analysis is impenetrable and unconvincing.

The authors could simplify their presentation greatly by stating accurately and transparently what they did analytically to develop circumstantial evidence implicating REST, without claiming either a lack of bias or some generalizable approach, and truncating the early figures correspondingly. Figure 5 could be dropped entirely.

We agree that the bio-informatic evidence is only one level of evidence, circumstantial or not, and must be followed up, which is what we have done. We strongly disagree with the reviewer’s opinion that whether REST was identified by this analysis is immaterial. Prior to embarking on many years

experimentation in multiple laboratories, we used transcriptomic data to query what differences we could observe between CNS and PNS using a combination of published algorithms. This led us to test RESTs function. Without a doubt, we would not have done so without those initial data, nor would our collaborators have agreed without some strong prior. These analyses provide such a prior, and its relevance supported by the comments of the other reviewers.

Following the reviewer's suggestions to present our analyses more transparently, we have removed 'unbiased' in our TF network analysis and changed the text to 'predicted REST binding sites' where TF footprinting analysis is involved. We also added more details in both the hierarchical network analysis and the footprinting analysis to be more clear in the Methods. As explained above, DNA footprinting of REST is an efficient approach to predict whether REST binds to the core regenerative TFs (ATF3, JUN, SMAD1, SOX11), a hypothesis borne from the network analysis. Our analysis indeed observed REST footprints on these genes, experimentally supporting direct binding. We therefore kept Fig.5 with the major revisions described.

Reviewer #3 (Remarks to the Author):

The authors have addressed my concerns, I have no further issues.

Thank you.

Reference:

- Sofroniew MV. Dissecting spinal cord regeneration. *Nature*. 2018 May;557(7705):343-350. doi: 10.1038/s41586-018-0068-4. Epub 2018 May 16. PMID: 29769671.
- Ma HW, Kumar B, Ditges U, Gunzer F, Buer J, Zeng AP. An extended transcriptional regulatory network of *Escherichia coli* and analysis of its hierarchical structure and network motifs. *Nucleic Acids Res*. 2004 Dec 16;32(22):6643-9. doi: 10.1093/nar/gkh1009. PMID: 15604458; PMCID: PMC545451.
- Ma HW, Buer J, Zeng AP. Hierarchical structure and modules in the *Escherichia coli* transcriptional regulatory network revealed by a new top-down approach. *BMC Bioinformatics*. 2004 Dec 16;5:199. doi: 10.1186/1471-2105-5-199. PMID: 15603590; PMCID: PMC544888.
- Balázsi G, Barabási AL, Oltvai ZN. Topological units of environmental signal processing in the transcriptional regulatory network of *Escherichia coli*. *Proc Natl Acad Sci U S A*. 2005 May 31;102(22):7841-6. doi: 10.1073/pnas.0500365102. Epub 2005 May 20. PMID: 15908506; PMCID: PMC1142363.
- Yu H, Gerstein M. Genomic analysis of the hierarchical structure of regulatory networks. *Proc Natl Acad Sci U S A*. 2006 Oct 3;103(40):14724-31. doi: 10.1073/pnas.0508637103. Epub 2006 Sep 26. PMID: 17003135; PMCID: PMC1595419.
- Bhardwaj, N., Yan, K. K., & Gerstein, M. B. (2010). Analysis of diverse regulatory networks in a hierarchical context shows consistent tendencies for collaboration in the middle levels. *Proceedings of the National Academy of Sciences*, 107(15), 6841-6846.
- Bhardwaj, N., Kim, P. M., & Gerstein, M. B. (2010). Rewiring of transcriptional regulatory networks: hierarchy, rather than connectivity, better reflects the importance of regulators. *Science signaling*, 3(146), ra79-ra79.
- Gerstein, M. B., Kundaje, A., Hariharan, M., Landt, S. G., Yan, K. K., Cheng, C., ... & Snyder, M. (2012). Architecture of the human regulatory network derived from ENCODE data. *Nature*, 489(7414), 91-100.
- Song L, Huang SC, Wise A, Castanon R, Nery JR, Chen H, Watanabe M, Thomas J, Bar-Joseph Z, Ecker JR. A transcription factor hierarchy defines an environmental stress response network. *Science*. 2016 Nov 4;354(6312):aag1550. doi: 10.1126/science.aag1550. PMID: 27811239; PMCID: PMC5217750.
- Alon, U. Network motifs: theory and experimental approaches. *Nature Rev. Genet.* 8, 450–461 (2007)
- Milo R, Shen-Orr S, Itzkovitz S, Kashtan N, Chklovskii D, Alon U. Network motifs: simple building blocks of complex networks. *Science*. 2002 Oct 25;298(5594):824-7. doi: 10.1126/science.298.5594.824. PMID: 12399590.
- Li Z, Schulz MH, Look T, Begemann M, Zenke M, Costa IG. Identification of transcription factor binding sites using ATAC-seq. *Genome Biol*. 2019 Feb 26;20(1):45. doi: 10.1186/s13059-019-1642-2. PMID: 30808370; PMCID: PMC6391789.
- M. Bentsen et al., ATAC-seq footprinting unravels kinetics of transcription factor binding during zygotic genome activation. *Nat Commun* 11, 4267 (2020).
- Vierstra J, Lazar J, Sandstrom R, Halow J, Lee K, Bates D, Diegel M, Dunn D, Neri F, Haugen E, Rynes E, Reynolds A, Nelson J, Johnson A, Frerker M, Buckley M, Kaul R, Meuleman W, Stamatoyannopoulos JA. Global reference mapping of human transcription factor footprints. *Nature*. 2020 Jul;583(7818):729-736. doi: 10.1038/s41586-020-2528-x. Epub 2020 Jul 29. PMID: 32728250; PMCID: PMC7410829.

Pierce SE, Granja JM, Greenleaf WJ. High-throughput single-cell chromatin accessibility CRISPR screens enable unbiased identification of regulatory networks in cancer. *Nat Commun.* 2021 May 20;12(1):2969. doi: 10.1038/s41467-021-23213-w. PMID: 34016988; PMCID: PMC8137922.

Cheng L, Li Y, Qi Q, Xu P, Feng R, Palmer L, Chen J, Wu R, Yee T, Zhang J, Yao Y, Sharma A, Hardison RC, Weiss MJ, Cheng Y. Single-nucleotide-level mapping of DNA regulatory elements that control fetal hemoglobin expression. *Nat Genet.* 2021 Jun;53(6):869-880. doi: 10.1038/s41588-021-00861-8. Epub 2021 May 6. PMID: 33958780; PMCID: PMC8628368.

Tian, F., Cheng, Y., Zhou, S., Wang, Q., Monavarfeshani, A., Gao, K., ... & He, Z. (2022). Core Transcription Programs Controlling Injury-Induced Neurodegeneration of Retinal Ganglion Cells. *bioRxiv*.

Reviewers' Comments:

Reviewer #2:

Remarks to the Author:

The authors have made some appropriate revisions. However, it remains puzzling to this reviewer that they continue to cling to two questionable features of their analysis when neither is necessary for their core results. One of these (networks) continues to be misguided and to distort or obscure the underlying biology, and the other (footprints) is fundamentally flawed and adds absolutely nothing to the story. The authors have not really changed anything in this version in response to prior critiques.

1) Networks

1.1) The issue here seems to me at a high level to be quite simple: The authors performed some set of analyses that suggested REST might be involved in certain biological processes. They then proceeded to perform various experiments showing that REST does in fact play an important role. But the mere fact of their results DOES NOT read on the soundness of their initial analyses. It is one thing to say that "we did a,b,c and got some hints which we then tested". It is quite another to say that "we have some robust procedure x that yielded result y" – with the implication that other researchers should try procedure x with the expectation that they too will get result y.

1.2) Regarding the network architecture, the authors make the strong claim that it is "unlikely we enriched REST-containing loops". I do not see how they can make this claim without supporting evidence. This is straightforward to test. The implication is that if they have in fact enriched for REST-containing loops then their entire analysis becomes tautological. If they stick to their guns, they need to address this issue one way or another. For clarity, my earlier comments were not referring to the ARACNE procedure, but rather to the forcing of the ARACNE network into a 3 layer structure – it is in the latter context that they need to show that they have not inadvertently enriched for REST-containing loops..

1.3) I also remain puzzled why the authors continue to cling to the concept of a 3-layer network architecture as being of core utility for the types of analyses they are performing. The 3-layer network structure is not some natural order of networks; rather, it is an analytical convenience that was expounded by a single laboratory (M. Gerstein) for the purpose of making certain global statements about classes of transcription factors and for making comparisons between species. By contrast, the network motifs (U. Alon and utilized by many others) are fundamental building blocks of regulatory network architectures that recur in diverse classes of naturally occurring architectures across species. The most prevalent and extensively analyzed of these structural motifs is the feed forward loop, which has the form $A \rightarrow B \rightarrow C$ with a parallel arm of $A \rightarrow C$. The authors could have drastically simplified and strengthened their analyses by simply focusing on the TFs that are found at the 3 different levels of feed forward loops.

1.4) The authors make the counterintuitive statement "this global structure [ie the 3-layer network] contains all possible formats of these small recurring regulatory patterns". The authors appear to be saying that their 3 layers still contain all of the classical network motifs – however they provide no evidence for this. If they did not in fact strip out certain kinds of network substructures then their network post 3-layer construction should have them all – they need to show this because a priori it does not seem tenable. Further, they need to show what is the relationship between their 3-layer network and classical feed-forward loops. Are the top nodes of the loops always in the top layer, etc?

1.5) Regarding quantitative differences between the PNS/CNS networks, the authors added representations of in/out degree for their various layers. However, if I am understanding them correctly, their figures pasted in-line in the response document raise new questions as CNS2/3 data sets appear to cluster in network layers 2 and 3 (the '2' and '3' labels having no relation to each other). Why is this the case?

2) Footprints

The authors still do not show evidence of any actual DNA footprinting. The citations they make simply obscure the issue because the cited papers also dodge this issue. DNA footprinting is fundamentally a nucleotide-level phenomenon. To claim that they actually have footprinting, they need to show nucleotide-level data. Their figure showing 'peaks' where footprints are supposedly occurring, with an x-axis reading out in kilobases, is not evidence that there are actual footprints in their data. Appeal to an algorithm that supposedly identifies footprints does not rescue them because the algorithm itself may be flawed. If there are footprints, they should show them or remove this section and the related claims.

Reviewer #2 (Remarks to the Author):

The authors have made some appropriate revisions. However, it remains puzzling to this reviewer that they continue to cling to two questionable features of their analysis when neither is necessary for their core results. One of these (networks) continues to be misguided and to distort or obscure the underlying biology, and the other (footprints) is fundamentally flawed and adds absolutely nothing to the story. The authors have not really changed anything in this version in response to prior critiques.

1) Networks

1.1) The issue here seems to me at a high level to be quite simple: The authors performed some set of analyses that suggested REST might be involved in certain biological processes. They then proceeded to perform various experiments showing that REST does in fact play an important role. But the mere fact of their results DOES NOT read on the soundness of their initial analyses. It is one thing to say that “we did a,b,c and got some hints which we then tested”. It is quite another to say that “we have some robust procedure x that yielded result y” – with the implication that other researchers should try procedure x with the expectation that they too will get result y.

Thank you for clarifying this point. To a large extent, we do agree with the Reviewer here. We haven't provided a “proof” that we have developed a robust pipeline that others should follow, we can only provide the evidence that we have garnered. We do demonstrate that REST is a major inhibitor of CNS regeneration. We show an example of using the network approach to organize high-dimensional data, and to prioritize hypotheses (which the reviewer calls hints) that could be tested in our animal models. We emphasize that this is not the case of a stopped clock being right twice a day, since there are multiple published successful examples of the general network approach being used by us and others to prioritize hypotheses for testing (Chandran et al., 2016; Swarup et al., 2019; Carro et al., 2010; Gatta et al., 2012). We are certainly not saying that this is always going to work, which would require a more solid theoretical basis for understanding biological processes (similar to physics or mathematics) that is simply not present (see also Geschwind and Konopka *Nature* 2009). We have softened language in the manuscript to align more with this reviewer's perspective.

1.2) Regarding the network architecture, the authors make the strong claim that it is “unlikely we enriched REST-containing loops”. I do not see how they can make this claim without supporting evidence. This is straightforward to test. The implication is that if they have in fact enriched for REST-containing loops then their entire analysis becomes tautological. If they stick to their guns, they need to address this issue one way or another. For clarity, my earlier comments were not referring to the ARACNE procedure, but rather to the forcing of the ARACNE network into a 3 layer structure – it is in the latter context that they need to show that they have not inadvertently enriched for REST-containing loops..

We thank the reviewer for clarifying their comment and agree with the Reviewer's comment that “because REST is known to control a large regulon, and it may be positioned as a superior node in many feed forward loops (FFLs)...”. If we were indeed biased to identify REST-containing loops by this confounder, we would expect to see REST enriched with FFLs within the networks. To test this, we performed the network motif analysis as suggested by the Reviewer (details in the response to the Reviewer's next comment, below). We do observe an enrichment of FFLs in the 3-layered PNS networks (**Reviewer Fig. 1, below**), but REST does not appear as a regulator in these FFL networks (Fig.1 C-D). REST appears in the CNS network, which does not enrich for FFLs, but adopts a simpler, bi-layered, and less inter-connected structure. This shows we are not inadvertently enriching for REST in FFLs.

1.3) I also remain puzzled why the authors continue to cling to the concept of a 3-layer network architecture as being of core utility for the types of analyses they are performing. The 3-layer network structure is not some natural order of networks; rather, it is an analytical convenience that was expounded by a single laboratory (M. Gerstein) for the purpose of making certain global statements about classes of transcription factors and for making comparisons between species. By contrast, the network motifs (U. Alon and utilized by many others) are fundamental building blocks of regulatory network architectures that recur in diverse classes of naturally occurring architectures across species. The most prevalent and extensively analyzed of these structural motifs is the feed forward loop, which has the form $A \rightarrow B \rightarrow C$ with a parallel arm of $A \rightarrow C$. The authors could have drastically simplified and strengthened their analyses by simply focusing on the TFs that are found at the 3 different levels of feed forward loops.

Again, we emphasize that without a clear proved theorem or theoretical model that fully explains transcriptional regulation and its relationship to biology more broadly, any architecture is just a model. We also agree that the concept of network motifs is extremely useful and seemingly more fundamental as a building block. So, as suggested by the Reviewer, we performed network motif analysis of each TF network using the mFinder software developed by U. Alon's group (Milo et al., 2002). Among the thirteen 3-node motif structures, we do indeed observe an enrichment of the FFLs in the PNS networks (**Reviewer Fig. 1A, below, motif ID 38, blue colored**). The CNS networks lack enrichment (or depletion) in most of the 3-node motif structures (**Reviewer Fig. 1A, red colored**), likely due to their simpler, bi-layered network structure (Fig. 1D).

Since the 3-layer network structure and the 3-node network motifs mainly occur in the PNS, we further explored the relationship between the two. We counted the frequency of each PNS node's appearance at each level, focusing on enriched network motifs with a clear top-down hierarchical structure (**Review Fig. 1A, motif ID 38,46,108**). We observed a general agreement in node hierarchy between the two structures. For example, JUN is a member of the 5-TF core sub-network and consistently appears as a top-node in the 3-layer PNS TF networks; It also prefers to occupy the top position in the network motif structure (**Review Fig. 1B**). Other core TF members including SOX11 and STAT3 prefer the middle-level position in both structures, while ATF3 sits at the bottom. We have included this analysis in the supplemental material.

1.4) The authors make the counterintuitive statement “this global structure [ie the 3-layer network] contains all possible formats of these small recurring regulatory patterns”. The authors appear to be saying that their 3 layers still contain all of the classical network motifs – however they provide no evidence for this. If they did not in fact strip out certain kinds of network substructures then their network post 3-layer construction should have them all – they need to show this because a priori it does not seem tenable.

To clarify, our statement was that the method to build the 3-layer network architecture is not biased to certain network motifs. We cited Jothi et al., 2009, Gerstein et al., 2012 to support this statement, in which they analyzed a comprehensive set of TFs in yeast or human, observing occurrence of many formats of network motifs within or across different layers. In our case, we focused on 21 core regenerative TFs discovered in our published work (Chandran et al., 2016), so our TF networks likely may not contain all possible formats and we are not making that claim, nor are our analyses dependent on it.

Further, they need to show what is the relationship between their 3-layer network and classical feed-forward loops. Are the top nodes of the loops always in the top layer, etc?

Please see our responses above. We found that the 3-layer network is in general agreement with the 3-node motif structures. JUN, EGR1, FOS and SP1, which appear as the top-nodes in the 3-

layer PNS networks, also prefer the top-level in the network motif structures (**Reviewer Fig. 1**).

1.5) Regarding quantitative differences between the PNS/CNS networks, the authors added representations of in/out degree for their various layers. However, if I am understanding them correctly, their figures pasted in-line in the response document raise new questions as CNS2/3 data sets appear to cluster in network layers 2 and 3 (the '2' and '3' labels having no relation to each other). Why is this the case?

In the CNS networks, most TFs form a two-level structure, with REST and CTCF interacting with top-layer TFs. We apologize for accidentally missing the full graph in the original response to the reviewer and have corrected that. We now show a revised version below, with CNS and PNS networks separated with correct labels.

2) Footprints

The authors still do not show evidence of any actual DNA footprinting. The citations they make simply obscure the issue because the cited papers also dodge this issue. DNA footprinting is fundamentally a nucleotide-level phenomenon. To claim that they actually have footprinting, they need to show nucleotide-level data. Their figure showing 'peaks' where footprints are supposedly occurring, with an x-axis reading out in kilobases, is not evidence that there are actual footprints in their data. Appeal to an algorithm that supposedly identifies footprints does not rescue them because the algorithm itself may be flawed. If there are footprints, they should show them or remove this section and the related claims.

We share with the Reviewer's concern that using ATAC-seq to footprint individual genomic site could be challenging, as many existing methods do not correct for Tn5 insertion bias. We used TOBIAS (Bentson et al., 2020), whose footprint calling algorithm uses a dinucleotide weight matrix to model and correct background bias of Tn5. TOBIAS identifies TF binding sites comparable to ChIP-seq data, and predicts TF footprints comparable to other popular tools including HINT-ATAC (Bentson et al., 2020), another *de novo* footprint caller effective to predict cleavage biases on both ATAC-seq and

DNase-seq data (Li et al., 2019). Since its recent publication, TOBIAS has been used and recommended by a number of published studies to effectively identify TF footprints, exactly as we do here (Bendl et al., 2021; Trevino et al., 2021; Sumida et al., 2021; Grandi et al., 2022). *Further, we must emphasize that we experimentally show that these foot-printing data overlap with previous REST binding sites identified in immature hippocampal neurons by ChIP (ref 53) and that the genes identified by foot-printing are significantly enriched in those that are changed with REST deletion (Fig 4G). Thus, we provide multiple layers of evidence that these genes are indeed transcriptionally regulated by REST binding.*

In the TOBIAS paper, the authors have shown nucleotide-level TF footprints (Bentson et al., 2020; Fig. 4 and supplemental Fig. 5). Here, we include an example of predicted REST footprint at the promoter region of *Stat3* (**Reviewer Fig. 3, below**), demonstrating uncorrected, expected Tn5 bias, and bias-corrected footprint signals, which were used to calculate footprint scores across multiple injury conditions, as shown in Fig.5C.

References:

- R. Jothi *et al.*, Genomic analysis reveals a tight link between transcription factor dynamics and regulatory network architecture. *Mol Syst Biol* **5**, 294 (2009).
- Gerstein, M. B., Kundaje, A., Hariharan, M., Landt, S. G., Yan, K. K., Cheng, C., ... & Snyder, M. (2012). Architecture of the human regulatory network derived from ENCODE data. *Nature*, 489(7414), 91-100.
- V. Chandran *et al.*, A Systems-Level Analysis of the Peripheral Nerve Intrinsic Axonal Growth Program. *Neuron* **89**, 956-970 (2016).
- Swarup, V., Hinz, F. I., Rexach, J. E., Noguchi, K. I., Toyoshiba, H., Oda, A., ... & Geschwind, D. H. (2019). Identification of evolutionarily conserved gene networks mediating neurodegenerative dementia. *Nature medicine*, 25(1), 152-164.
- M. S. Carro *et al.*, The transcriptional network for mesenchymal transformation of brain tumours. *Nature* **463**, 318-325 (2010).
- Geschwind DH, Konopka G. Neuroscience in the era of functional genomics and systems biology. *Nature*. **461**: 908-15 (2009).
- G. Della Gatta *et al.*, Reverse engineering of TLX oncogenic transcriptional networks identifies RUNX1 as tumor suppressor in T-ALL. *Nature medicine* **18**, 436-440 (2012).
- Alon, U. Network motifs: theory and experimental approaches. *Nature Rev. Genet.* **8**, 450–461 (2007)
- Milo R, Shen-Orr S, Itzkovitz S, Kashtan N, Chklovskii D, Alon U. Network motifs: simple building blocks of complex networks. *Science*. 2002 Oct 25;298(5594):824-7. doi: 10.1126/science.298.5594.824. PMID: 12399590.
- M. Bentsen *et al.*, ATAC-seq footprinting unravels kinetics of transcription factor binding during zygotic genome activation. *Nat Commun* **11**, 4267 (2020).
- Li, Z. et al. Identification of transcription factor binding sites using ATAC-seq. *Genome Biol.* **20**, 45–45 (2019).
- Bendl, J., Hauberg, M. E., Girdhar, K., Im, E., Vicari, J. M., Rahman, S., ... & Roussos, P. (2021). The three-dimensional landscape of chromatin accessibility in Alzheimer's disease. *BioRxiv*.
- Trevino, A. E., Müller, F., Andersen, J., Sundaram, L., Kathiria, A., Shcherbina, A., ... & Greenleaf, W. J. (2021). Chromatin and gene-regulatory dynamics of the developing human cerebral cortex at single-cell resolution. *Cell*, 184(19), 5053-5069.
- Sumida, T. S., Dulberg, S., Schupp, J. C., Lincoln, M. R., Stillwell, H. A., Axisa, P. P., ... & Hafler, D. A. (2022). Type I interferon transcriptional network regulates expression of coinhibitory receptors in human T cells. *Nature Immunology*, 1-11.
- Grandi, F. C., Modi, H., Kampman, L., & Corces, M. R. (2022). Chromatin accessibility profiling by ATAC-seq. *Nature Protocols*, 1-35.